# Effects of nitrogen and phosphorus amendments on $CO_2$ and $CH_4$ production in peat soils of Scotty Creek, Northwest Territories: Potential considerations for wildfire and permafrost thaw impacts on peatland carbon exchanges

Eunji Byun[1], Fereidoun Rezanezhad[2], Stephanie Slowinski[2], Christina Lam[2], Saraswati Bhusal[2,3], Stephanie Wright[4], William L. Quinton[5], Kara L. Webster[6], Philippe van Cappellen[2]

[1]Department of Earth System Sciences, Yonsei University, Seoul, 03722, Korea
[2]Ecohydrology Research Group, Water Institute and Department of Earth and Environmental Sciences, University of Waterloo, Waterloo, Ontario, N2L 3G1, Canada
[3]Department of Geography and Geospatial Sciences, South Dakota State University, Brookings, SD 57007, USA
[4]Department of Civil Engineering, Queen's University, Kingston, ON K7L 3N6, Canada
[5]Cold Regions Research Centre, Wilfrid Laurier University, Waterloo, ON N2L 3C5, Canada
[6]Great Lakes Forestry Centre, Canadian Forest Service, Natural Resources Canada, Sault Ste. Marie, ON P6A 2E5, Canada

*Correspondence to*: Eunji Byun (eb@yonsei.ac.kr)

**Abstract.** Impacts of nutrient enrichment on soil carbon cycling have been extensively studied in temperate and tropical regions where intensive agriculture and land development have led to large increases in anthropogenic inputs of nitrogen (N) and phosphorous (P). However, how soil carbon sequestration and soil-atmosphere gas exchanges in cold regions respond to greater inputs of N and P remains poorly known, despite recent observations showing significant increases in porewater N and P in burned subarctic peatlands and downstream waters. Wildfires plus enhanced hydrological connectivity due to permafrost thaw have therefore the potential to change carbon turnover and gas emissions in soils of northern peatlands. To start exploring the sensitivity of peatland soil biogeochemistry to variations in N and P availability, we measured the carbon dioxide ($CO_2$) and methane ($CH_4$) production rates during a month-long incubation experiment with soils from a bog and fen collected at the long-term Scotty Creek research station in the Northwest Territories, Canada. Subsamples of the peatland soils were divided into containers to which artificial porewater solutions were added. These solutions were amended with either dissolved inorganic N, dissolved inorganic P, or dissolved N and P together. Unamended controls were run in parallel. The containers cycled through pre-set temperature steps of 1, 5, 15, and 25°C. Overall, the fen soil yielded higher $CO_2$ and $CH_4$ production rates than the bog soil. Amendment of N to the bog soil produced more $CO_2$ compared to its control, while amendment of P increased $CO_2$ production in the fen soil. Amendment of N and P together reduced $CO_2$ production but increased that of $CH_4$ in both the fen and bog soil incubations. Porewater chemistry at the end of the 30-day experiment showed aqueous C, N, and P stoichiometric ratios that trended toward those of the soil microbial biomasses, hence, implying that the initial microbial nutrient status played a crucial role in determining the responses to the different nutrient amendments. Our results demonstrate that porewater nutrient availability and soil carbon cycling interact in complex ways to change $CO_2$ and $CH_4$ production rates

in peatland soils, with potentially far-reaching implications for the impacts of wildfires and permafrost thaw on peatland-atmosphere carbon exchanges.

## 1 Introduction

Thawing permafrost and collapsing peatlands pose a threat to the stability of the net ecosystem carbon sequestration in subarctic regions (Treat et al., 2019; Schuur et al., 2022). While permafrost thaw and peatland collapse is rapidly expanding across these regions (Porter et al., 2019; Quinton et al., 2019; Hugelius et al., 2020), some parts of the northern boreal and subarctic regions, such as western Canada (Gibson et al., 2018), Siberia (Talucci et al., 2022), and Alaska (Mekonnen et al., 2022), have also experienced increased wildfire activity in recent years likely further altering the effects of permafrost thaw on soil carbon stability. For example, increases in the number and extent of fire events result not only in the immediate carbon loss through biomass burning but also an increase of downstream export of particulate and dissolved organic carbon (DOC) from the burned areas (Burd et al., 2018; Burd et al., 2020; Ackley et al., 2021; Koch et al., 2022; Mekonnen et al., 2022).

Increasing N and P inputs to peatland soils have the potential to perturb soil carbon cycling in peat soils. For instance, by relieving nutrient limitations on the soil microbial community, additional N and P may accelerate the decomposition of soil organic matter. Recent observations reported that the surface water and shallow groundwater in burned areas were substantially enriched in dissolved nitrogen (N) and phosphorus (P), with high P concentrations possibly persisting for several years after a fire (Emelko et al., 2016; Van Beest et al., 2019; Emmerton et al., 2020; Orlova et al., 2020). Thawing permafrost may release additional dissolved nutrients previously bound in frozen sediments (Treat et al., 2019; Schuur et al., 2022; Wright et al., 2022). Furthermore, at the landscape level, permafrost thaw-induced ground subsidence increases the hydrological and geochemical connectivity among landscape components, including forested areas, fens, thermokarst bogs, and adjacent peatlands (Connon et al., 2014; Gibson et al., 2018; Haynes et al., 2018; Post et al., 2019; Carpino et al., 2021). Thus, there is growing probability for organic carbon stored in peat soils to experience enhanced microbial decomposition driven by nutrient enrichment.

While peatland soils tend to be generally poor in nutrients such as N and P, the magnitude of nutrient limitation of the soil microbial community varies with landscape position, peatland type, and groundwater connectivity (Hill et al., 2014; Lin et al., 2014; Moore et al., 2019). For example, isolated ombrotrophic bogs rely on direct atmospheric deposition of N and P plus microbial N fixation. In comparison, minerotrophic channel fens capture nutrients from surrounding water bodies through groundwater and surface water pathways. Rapid increases in nutrient inputs to peatlands have long been studied in temperate and tropical regions, where there is a higher chance of fertilizer spillover and growing agriculture or fossil fuel-driven atmosphereic deposition (Amador and Jones, 1993; Hoyos-Santillan et al. 2018; Lin et al., 2014; Moore et al., 2019; Qualls and Richardson, 2000; Schillereff et al., 2021). Early research on subtropical peatlands in the Everglades, Florida, showed that

experimentally P-enriched soils released significantly more $CO_2$ than controls, seemingly because the added nutrient P stimulated soil heterotrophic respiration (Amador and Jones, 1993; Qualls and Richardson, 2000). Recently, however, a study suggested that nutrient availability is not always a rate-determining factor as microbial communities in tropical peatland soils have developed strategies to process site-specific plant litter even without adequate nutrient supply (Hoyos-Santillan et al., 2018). Still, other studies support the important control of nutrient increases in peat decomposition rates. For example, a data

synthesis across temperate regional bogs revealed that P supply to surface peat enhanced microbial decomposition activities and reduced net organic C burial to deeper peat layers (Schillereff et al., 2021). Another study of a 12-year P fertilization field experiment for temperate mountainous peatlands showed that the extra P supply increased soil respiration and $CO_2$ release, reducing the overall carbon sink function of the peatland ecosystem there (Lu et al., 2022).

Given the high accumulation of plant organic material in peatlands, the absolute amount of carbon as substrate for microbial activities is assumed not limiting in peat soils. Microbial strategies to overcome relative nutrient imbalances of N, P or other nutrients relative to carbon may fall into three broad categories: (1) increasing the production of extracellular enzymes to acquire necessary nutrient elements from enzyme-facilitated organic matter degradation, (2) recycling of the nutrient elements assimilated in dead microbial cells (i.e., necromass recycling), and (3) releasing some extra carbon in relation to the other

major nutrients via auxiliary respiration without biomass assimilation (i.e., overflow respiration) (Giesler et al., 2011; Manzoni et al., 2012; Lin et al., 2014; Hoyos-Santillan et al., 2018; Schillereff et al., 2021; Lu et al., 2022). To address which strategy is activated, soil incubation experiments provide one approach to measure the changes in carbon gas production rates and net microbial biomass changes.

The Scotty Creek watershed, Northwest Territories, Canada, hosts one of the main research stations in the subarctic region, with active monitoring and field investigations on rapidly changing discontinuous permafrost landscapes and potential biogeochemical climate feedbacks (http://www.scottycreek.com/). Recent observations highlighted the increase of wildfire impacts, the acceleration of permafrost thaw and the degradation of permafrost stability As a result, peatland ecosystems and their associated hydrological processes and pathways are rapidly transforming (Gibson et al., 2018; Ackley et al., 2021; Wright

et al., 2022). However, the potential effects of sudden nutrient inputs to peatland soils have not been explored for subarctic regions, despite the possible increase of organic carbon decomposition and greenhouse gas emissions suggested by studies in other climate zones. In this study, our aim was to investigate the potential acceleration of organic matter decomposition and carbon gas production in subarctic peatland soils following the addition of readily bioavailable dissolved inorganic N and P. Also, we anticipated that variations in initial nutrient ratios (N:P, C:N, C:P) and the existing adaptations of soil microbial

communities would contribute to understanding the short-term responses of different types of peatlands to additional N or P (Hill et al., 2014; Hoyos-Santillan et al., 2018). A laboratory soil incubation experiment was conducted to compare the $CO_2$ and $CH_4$ production rates in the field-sampled bog or fen soils under various treatments: control, N only, P only, and both N

and P (NP hereafter) addition. Fluctuating temperatures were imposed during the month-long incubation to analyze the temperature sensitivity of the gas production rates in the different nutrient treatments.

## 2 Methods

### 2.1 Scotty Creek field sites and peat coring

The sampling locations were within the Scotty Creek drainage basin, Northwest Territories, Canada, which lies within the Taiga Plains Ecozone (Fig. 1). Approximately 25% of this ecozone is covered by wetlands (Mahdianpari et al., 2021) with an estimate of 71,600 $km^2$ bog and 5,100 $km^2$ fen peatlands (Webster et al., 2018). The Scotty Creek basin is underlain by discontinuous permafrost that supports a peatland complex with peat plateau forests, thermokarst bogs, channel fens, and open water ponds. In October 2020, two duplicate shallow peat cores (0-25 cm) were taken from each of two sites: a thermokarst bog covered by *Sphagnum* mosses (hereafter referred to as 'bog peat' or 'bog soil') and a channel fen covered by herbaceous plants (hereafter referred to as 'fen peat' or 'fen soil') as shown in Fig. 1. The peat cores were collected in transparent plastic liners (3-inch inner diameter, AMS, Inc). These cores were transported to the University of Waterloo, Waterloo, Canada, and then stored in a -20°C freezer until being thawed to start the experiment.

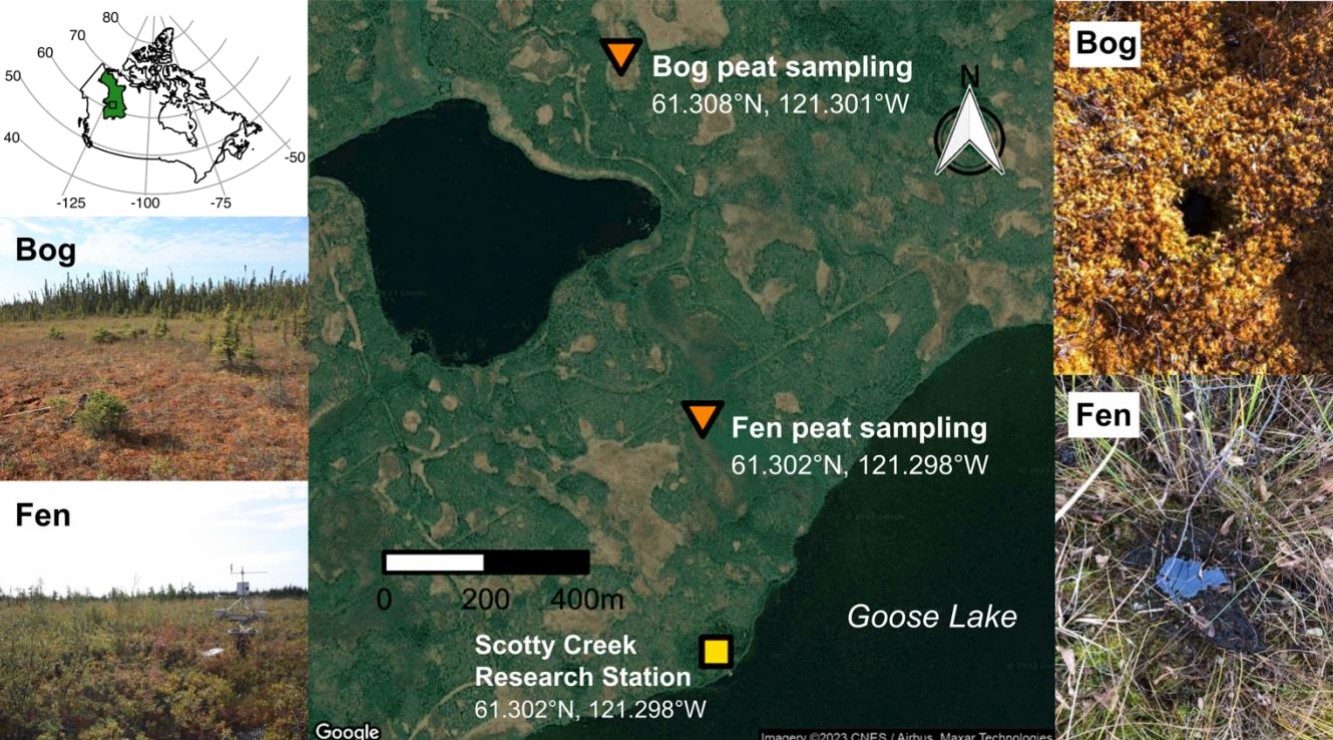

**Figure 1: Scotty Creek watershed bog and fen peat sampling site locations (Map data: Google ©2023 CNES / Airbus, Maxar Technologies) and photos (credit: Mason Dominico). The location and extent of the Taiga Plains Ecozone of Canada (National Ecological Framework for Canada, 2017) is shown in the inset map with a small box indicator for the site locations.**

## 2.2 Sample preparation and pre-incubation

The frozen peat cores were thawed at room temperature (22˚C). During the thawing, each peat core was sealed inside its original liner (from the field sampling) to preserve initial porewater released upon its thawing. After 24 hours, the liners were opened carefully, the released porewater from each core was collected and aliquots were saved for chemical analyses. The top layer containing fresh, large plant debris was removed from each peat core. The rest of the wet peat was placed in a clean plastic container and gently hand mixed to take small aliquots for determining approximate values of initial peat bulk density (approximate as the peat was de-structured), and moisture content (by oven-drying at 80˚C for 48 hours). In addition, a sub-sample of each of the peat soils was freeze-dried for analysis of the total organic carbon and total nitrogen and total phosphorus concentrations. Total organic carbon (TOC) and total nitrogen (TN) were determined using a CHNS analyzer (Carlo Erba NA-1500 Elemental Analyzer; detection limit of 1% by mass for both TOC and TN). Total phosphorus (TP) was determined following the method of Aspila et al. (1976): an aliquot of soil was burned to create ash in a muffle furnace at 550˚C with magnesium nitrate added as an oxidant. The ashed soil was subsequently extracted by mixing with 1 M hydrochloric acid on a shaker for 16 hours. The extract was then analyzed for total dissolved P by Inductively Coupled Plasma – Optical Emission Spectrometry (ICP-OES, Thermo Scientific iCAP 6300 Duo).

The containers with bulk peat samples were placed inside an environmental chamber at a constant temperature of 5˚C for two weeks. During this pre-incubation, the peat containers were uncovered inside the chamber, allowing for aerobic conditions. The pre-incubation caused some reduction of the peat moisture contents. We therefore re-measured the moisture contents at the end of the pre-incubation period by the oven-drying method to determine how much porewater solutions needed to be added to return the peat moisture contents to the selected levels. Note that the latter were selected to be close to average field conditions.

After the pre-incubation, each of the bog and fen peat samples was subsampled for fumigation to measure microbial biomass C, N and P concentrations. These concentrations were considered the baseline values against which changes in microbial C, N, and P accompanying the various nutrient treatments were evaluated. Then, the rest of the bulk peat was allocated into 250 mL mason glass jars (bog: 55 g peat and fen: 40 g peat). Porewater base solutions were prepared by diluting the initial thaw-collected porewater around 10:1, to make the required volume of the base solution while accounting for pH and initial nutrient levels, rather than using pure water. The moisture conditions for the incubation experiment were selected using field-based growing-season averages measured in adjacent meteorological stations equipped with soil sensors. The selected moisture conditions were a volumetric water content of 80% for the bog peat and 100% for the fen peat. These values were then used to calculate the volumes of artificial porewater to add to the experimental containers.

The artificial porewater solutions were prepared to adjust the peat moisture content and different treatments by adding different amounts of salts including monobasic sodium phosphate ($NaH_2PO_4$), ammonium nitrate ($NH_4NO_3$), and ammonium chloride ($NH_4Cl$) for the N and/or P porewater enrichment scenarios. This experiment was aimed to test high levels of porewater

nitrogen (7 mg-N $L^{-1}$) and phosphorus (6 mg-P L-1) from previous study measurements (Table 1). For the N amendment, the mix of $NH_4NO_3$ and $NH_4Cl$ was used to introduce a 3:1 ratio for $NH_4$-N to $NO_3$-N following the measurement from a burned peat surface (Table 1). The solutions were added to bog and fen peat samples for control, N added, P added, and both N and P added, all prepared in duplicate (4 treatments × 2 sites × 2 samples = 16 total incubation jars). After adding the solutions, the headspace was 100 mL for bog samples and 150 mL for fen samples. The jars were placed back in the environmental chamber

at 5˚C and pre-incubated for four weeks until stable $CO_2$ efflux rates were measured.

**Table 1: Nitrogen (N) and phosphorus (P) concentrations in wildfire-impacted water samples from recent studies in western boreal and subarctic Canada.**

| Site | Disturbance | Sampling and analysis methods | N reported (mg $L^{-1}$) | P reported (mg $L^{-1}$) | Reference |
|---|---|---|---|---|---|
| Scotty Creek and Notawohka Creek catchments, NWT | Wildfire 3 years ago (the 2013 Notawohka fire) | Burned catchment downstream water sampling | 0.5-0.9 | 0.06 | Burd et al. (2018) |
| Scotty Creek and Notawohka Creek catchments, NWT | Wildfire 3 years ago (the 2013 Notawohka fire) | Burned peatland porewater field sampling and light exposal incubations | 1.4-2.8 | 0.6 | Burd et al. (2020) |
| Fort McMurray, Alberta | Wildfire in the same year (the May 2016 Fort McMurray wildfire) | Surface water samples collected from high- to low-order rivers | 2-3 | 0.5-1 | Emmerton et al. (2020) |
| Pelican Mountain, Alberta | Prescribed burn at research site in the same year | Shallow groundwater samples from monitoring wells | 1-3 | 0.7 | Orlova et al. (2020) |
| Scotty Creek watershed, NWT | Wildfire 2 years ago (2014 low-severity fire in the headwater areas) | Burned peatland porewater, field collected; comparison with unburned area | 1-7 | 0.5-6 | Ackley et al. (2021) |
| Pelican Mountain, Alberta | Laboratory simulated burning of research site samples at 250 and 300°C | 5 g of burned peat leached by 1 litre water for 2 days | 220-420 | 12-26 | Wu et al. (2022) |

**2.3. Incubation experiment and subsampling**

The incubation jars were left open inside the environmental chamber to maintain aerobic headspace conditions, except during the closed headspace gas sampling. To maintain the moisture contents in the peat samples, the jars were placed together in a partially opened plastic bag with a small beaker containing a Milli-Q water-soaked sponge. The potential evaporative water loss was monitored by measuring the weights of two additional jars containing the same quantity of bog and fen peat, going

through the same temperature course. Using this information, the peat moisture content was maintained within 10% of the

initial condition by spraying some Milli-Q water inside the plastic bags to compensate for evaporation loss throughout the incubation experiment.

During the one-month incubation experiment, the headspace gas sampling was performed every third day after the chamber was set to a new temperature. Stepwise changes in the chamber temperature from 1˚C, 5˚C, 15˚C, 25˚C, 15˚C, 25˚C, 15˚C, 5˚C, 1˚C, 5˚C and back to 1˚C were imposed every 3-day interval to roughly mimic field situations where the ground temperature fluctuates during the non-growing to growing season transition. This non-growing to growing season transition, which includes snow melt, was thought to be a plausible scenario for the sudden contact of nutrient enriched water for distant peatlands after a watershed fire event. The comparison of flux rates at various temperatures was also aimed to test potential variation in nutrient interaction with microbial activities.

### 2.3.1 Porewater chemistry

Approximately 20 mL of porewater was extracted from bog and fen peat samples during the pre-incubation phase, and from each incubation jar during the post-incubation phase. The chemical properties of the porewater collected from the initial thawing of the peat cores were used for determining the initial porewater conditions and preparation of the artificial porewater solutions. The post-incubation porewater was collected by gently squeezing the saturated peat inside each incubation jar. The porewater pH was measured by a calibrated electrode (Orion™ Economy Series pH Combination Electrode, ThermoScientific) before filtering. The rest of the porewater samples were filtered through a 0.45 µm pore size membrane filter (nylon membrane syringe filters, VWR Scientific) and stored at 4˚C in the refrigerator if not analyzed immediately.

Concentrations of dissolved organic carbon (DOC), total dissolved nitrogen (TDN), and dissolved inorganic carbon (DIC) in the filtered porewater were measured using a total organic carbon analyzer (TOC-LCPH/CPN, Shimadzu; method detection limit 3 µmol $L^{-1}$). For the measurement, 1 mL of porewater was filtered through a 0.2 µm pore size membrane filter (Polyethersulfone membrane syringe filters, Thermo Scientific) and frozen for subsequent analysis of major anion concentrations including chloride ($Cl^-$), nitrite ($NO_2^-$), nitrate ($NO_3^-$), and sulfate ($SO_4^{2-}$) using ion chromatography (IC, Dionex ICS-5000 with a capillary IonPac® AS18 column; ± 3.0% error and ± 1.6% precision; method detection limit 0.59, 1.29, 1.13, 1.47 µmol L-1, respectively ). Porewater ammonium ($NH_4^+$) and dissolved reactive P (DRP) concentrations were measured spectrophotometrically on a Thermo Scientific™ Gallery™ Discrete Analyzer (±10% error and ±3% precision). The concentrations of major cations including dissolved calcium, iron, potassium, magnesium, manganese, sodium, sulfur, and silicon concentrations were measured using ICP-OES (Thermo Scientific iCAP 6300 Duo).

### 2.3.2. Microbial biomass fumigation

The microbial biomass carbon (MBC), nitrogen (MBN) and phosphorus (MBP) concentrations of the peat samples were measured using the chloroform fumigation method (Brookes et al., 1984; Vance et al., 1987; Joergensen, 1996; Jenkinson et al., 2004). For both the extractions of biomass P and biomass C and N, 4 g subsamples of the peat were sampled in parallel. One set of subsamples was treated with chloroform for 24 hours to fumigate the biomass in a vacuum desiccator, while a parallel set of subsamples was not fumigated. Both the fumigated and non-fumigated peat samples were then extracted for OC and N using 0.5 M potassium sulfate and for P using 0.5 M sodium bicarbonate ($NaHCO_3$ solution with pH adjusted to 8.5). Both the potassium sulfate extracts and the sodium bicarbonate extracts were then filtered through a 0.45 µm pore size membrane filter (nylon membrane syringe filters, VWR Scientific), and the extracts analyzed, respectively, for DOC and TDN by TOC-L and for total dissolved phosphorus by ICP-OES. The difference in DOC, TDN, and TDP concentrations between the fumigated and non-fumigated samples represented the C, N, and P present in the microbial biomass. Extraction efficiencies of 0.45, 0.54 and 0.4 were assumed for MBC, MBN and MBP, respectively (Joergensen, 1996) (Jenkinson et al., 2004). The resulting values from these sample measurements were compiled into a calculation spreadsheet to derive MBC, MBN and MBP values for each peat sample (Byun et al., 2024).

### 2.3.3. Headspace $CH_4$ and $CO_2$ flux measurements

The changes in $CO_2$ and $CH_4$ concentrations in the headspace of the jars during the incubation were measured by closing the jars for 15 or 20 minutes. Up to 10 mL headspace gas samples were taken from each jar at the end of the closed incubation through the custom lid with three-way gastight valves and a 10- or 20-mL plastic syringe. The $CO_2$ and $CH_4$ concentrations were analyzed by direct sample injection into a Gas Chromatograph (Shimadzu, Model GC-2014) equipped with a flame ionization detector and methanizer. The method detection limits for $CH_4$ and $CO_2$ were 0.384 and 17.145 ppm, respectively. For gas samples with $CO_2$ concentrations greater than 1,000 ppm, the samples were diluted with 15-30 mL of helium gas and allowed to mix for 20 minutes before analysis on the Gas Chromatograph. Gas efflux rates ($F_{gas}$, µmol g-1 h-1) were calculated following Eq. (1):

$$F_{gas} = \frac{PV_H(C_{t1}-C_{t0})}{RTm\Delta t} \tag{1}$$

where $P$ is the headspace pressure (atm), $V_H$ is the headspace volume (L), $R$ is the gas constant (0.0821 L atm mol$^{-1}$ K$^{-1}$) and $T$ is the temperature (K). Gas efflux was estimated using the relative headspace concentration changes inside the closed jar environments, as $C_{t1}-C_{t0}$ (ppm or 10-6) for each gas species during the time, $\Delta t=t_1-t_0$ (h). The gas efflux rates were then normalized per dry mass of peat ($m$, g dry peat) determined at the beginning of the incubation experiment (the mass loss was assumed minor and not considered in the flux rate calculations).

Based on the average gas production rates at 5°C and 25°C, the temperature sensitivity parameter $Q_{10}$ was calculated using an exponential rate increase following Eq. (2)

$$Q_{10} = (\frac{R2}{R1})^{(\frac{10}{T2-T1})} \tag{2}$$

where $R2$ is the rate at the incubation temperature 25°C ($T2$) and $R1$ at the 5°C ($T1$). However, the $CH_4$ production rates in the bog peat containers were very low and yielded too little variation to derive meaningful $Q_{10}$ values for $CH_4$ production.

## 3 Results and Discussion

### 3.1 Transient changes in the soil $CO_2$ and $CH_4$ production

The fen soil incubation resulted in higher $CO_2$ and $CH_4$ production rates than the bog soil incubation as the incubation temperature was cycling from 1 to 25°C (Fig. 2). The $CO_2$ production rates of the bog samples were about 50% of those of the fen samples and the $CH_4$ production rates of the bog samples were an order of magnitude smaller than those of the fen samples. The higher $CO_2$ and $CH_4$ production rates in the fen soil are consistent with the higher initial biomass of the fen soil (MBC 4.3 mg C g$^{-1}$) compared to that of the bog soil (MBC 0.7 mg C g$^{-1}$). Not accounting for primary production from living plants in

this experiment, this result of higher fen $CO_2$ production rates are inconsistent with previous Soctty Creek field measurements by Chasmer et al. (2012) which showed higher net $CO_2$ emissions in bogs than in fens.

Temperature variably affected the gas production rates in the different nutrient treatments. The bog peat soil with added P and NP (thus, P amendment effects in general) yielded distinctively lower $CO_2$ production rates at 25°C but higher $CO_2$ production

rates at 15°C, compared to the no-nutrient control (Fig. 2a). The fen peat soil with added N and NP (thus, N amendment effects in general) resulted in lower $CO_2$ production rates at 25°C but no apparent difference at 15°C, compared to the no-nutrient control. For the fen peat soil, the P only amendment (i.e., without N) resulted in distinctively higher $CO_2$ rates than the N amended treatments (N only and NP) and the control (Fig. 2b). The $CH_4$ production rates of the fen soil also showed noticeable temperature effects with the greatest differences between the various nutrient treatments observed at 25°C (Fig. 2b).

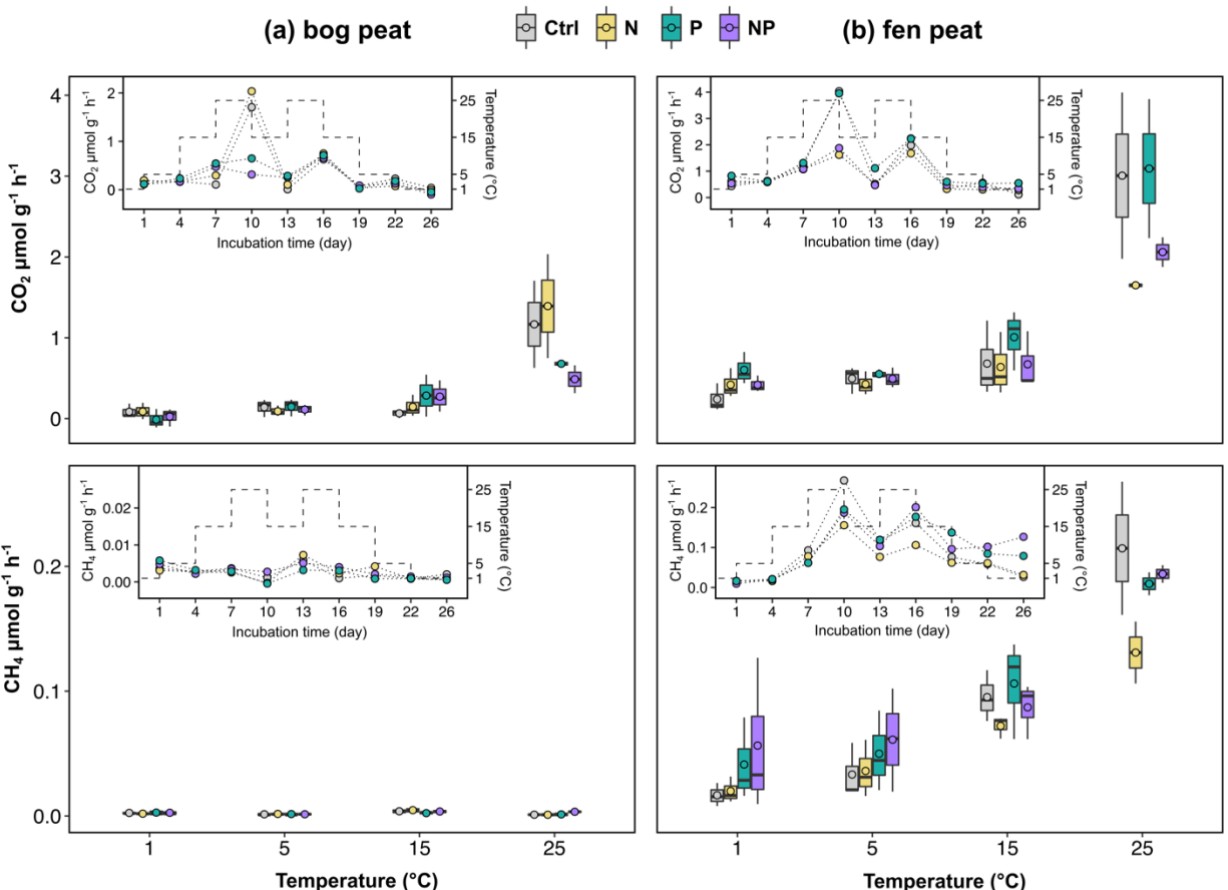

**Figure 2: The CO₂ and CH₄ flux rates from the laboratory incubation measurements for subarctic (a) bog and (b) fen peat samples. Inset figures show the rate changes (connected dots) for each nutrient treatment (Ctrl: control; N: nitrogen addition; P: phosphorus addition; NP: nitrogen and phosphorus addition) over the incubation period with temperatures shown as dashed step lines. Box plots show the summary of rates at each temperature (median line and mean circle in each box) for each treatment.**

In a broad sense, the $CO_2$ and $CH_4$ production rates followed the anticipated exponential increase with temperature (Fig. 2). The estimated $Q_{10}$ values suggest that the additional nutrients reduced the temperature sensitivity of the $CO_2$ emissions (i.e., the $Q_{10}$ values were lower), except for N addition to the bog soil (Table 2). According to the flux rate summary in Fig. 2, the low $Q_{10}$ values correspond to relatively slow flux rate increases with temperature, while the higher $Q_{10}$ represents the relatively fast flux rate increase with temperature rise to 25°C. For example, the NP addition resulted in the lowest $CO_2$ production rates in the bog soil at 25°C, contributing to the apparently lowest $Q_{10}$ of 2.09 (Table 2); in contrast, the N addition resulted in the highest $CO_2$ production rate in the bog soil at 25°C, contributing to the highest $Q_{10}$ of 3.89 (Table 2).

**Table 2: Estimated temperature sensitivity based on the average gas production rates at 5°C and 25°C.**

| Treatments | Bog $CO_2$ | Fen $CO_2$ | Fen $CH_4$ |
|---|---|---|---|
| Control | 2.9 | 2.5 | 2.5 |
| N addition | 3.9 | 2.0 | 1.9 |
| P addition | 2.2 | 2.4 | 1.9 |
| N P addition | 2.1 | 2.0 | 1.8 |

The temperature sensitivities estimated here suggest that the alleviation of nutrient limitation may result in either relatively high $CO_2$ production rates at the lower temperatures tested (according to low $Q_{10}$ values for both NP, bog P, and fen N), or high $CO_2$ production rates at the higher-end temperatures (according to the high $Q_{10}$ for bog N and fen P), or some combination of both trends. Recently, Liu et al. (2022) suggested that the catalytic efficiency of hydrolytic enzymes involved in soil N and P recycling significantly increases with increasing temperature. This would imply that microbial respiration at warmer temperature would become less dependent on the initial soil and N and P stores. However, as shown by our results, the coupled effects of nutrient limitation and temperature may be more complex than currently recognized. A better predictive understanding of the variations in $Q_{10}$ values across peatland classes remains a crucial task for the calibration of process-based C cycle models of peatlands (Bona et al., 2020). For example, the recently observed latitudinal increase of $Q_{10}$ values for $CO_2$ production in peat soils (Byun et al., 2021) could point to an increasing severity of nutrient limitation for peatlands at high latitudes.

## 3.2 Cumulative carbon release by nutrient addition

The stoichiometric ratios of primarily accessible substrates are important for soil microbial C cycling. The nutrient element ratios in the soil solution relative to the microbial biomass ratios can determine the use efficiency of DOC for respiration or biomass storage. This process has been conceptualized as microbial carbon use efficiency (CUE), which is expressed by the fraction of substrate OC that is taken up into microbial biomass of the total carbon that is used for both biomass uptake and cell metabolism (Manzoni et al., 2012; Sinsabaugh et al., 2013; Sinsabaugh et al., 2016; Geyer et al., 2016). If microbial cells are substantially limited by available nutrient N and/or P, they may allocate less of the DOC for biomass growth but still respire more (producing more $CO_2$ from DOC) for alleviating the nutrient limitation (thus, low CUE). To discuss this effect, the comparison chart for the cumulative emission of $CO_2$, $CH_4$ and the total of both gases (Gas C) is shown in Fig. 3, as the cumulative emissions were considered more indicative of the resulting net changes by the initial nutrient additions in peat carbon cycling over incubation.

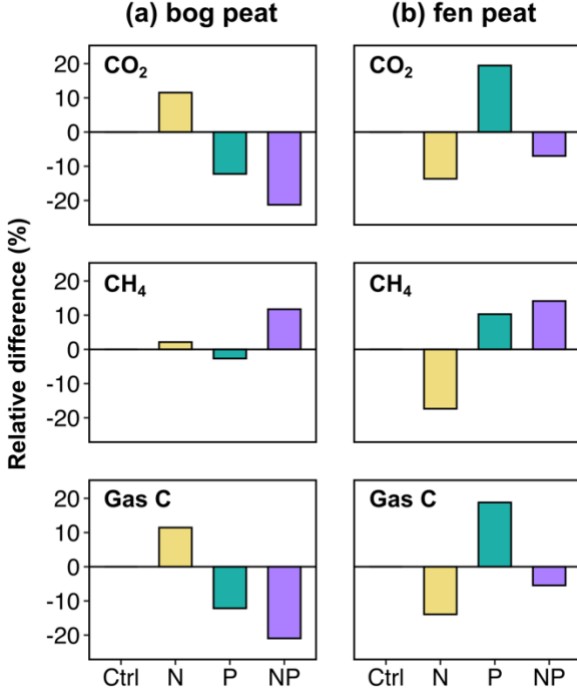

**Figure 3: Relative differences (%) in cumulative $CO_2$, $CH_4$, and total carbon gas (Gas C) emissions integrated over the entire incubation experiment: comparing the unamended control (Ctrl) and the nutrient amended treatments N (N only), P (P only) and NP (both N and P together).**

The microbial CUE was estimated for the bog soil by fumigation extraction method at the end of the incubation (Table 3). The fen soil fumigation experiment was not successful yielding negative microbial biomass calculations based on the sample weight and concentration measurements (Byun et al. 2024). The inferred negative biomass for the fen soil was considered a methodological artefact. Thus, only the microbial CUE of bog soil was approximated by the gross MBC acquisition, divided by the total of the MBC acquisition and the cumulative gas C release. As a result, the lower CUE by the N addition was attributed to the largest C gas release with the smallest gross MBC acquisition. The highest CUE value with the larger MBC acquisition and the smaller cumulative C gas release was found in the case of NP addition. This partially corresponds to the above interpretation of temperature sensitivity trend that the microbial respiration activities increased to alleviate relatively lacking nutrient P in the bog soil in response to the imbalance intensified by N addition.

**Table 3: Microbial biomass changes and carbon use efficiency (CUE) of bog peat.**

| Extraction | Before | | | After | | | | |
|---|---|---|---|---|---|---|---|---|
| Bog | MBC (mg C g⁻¹ dry soil) | MBN (mg N g⁻¹ dry soil) | MBC:MBN (atomic ratio) | MBC (mg C g⁻¹ dry soil) | MBN (mg N g⁻¹ dry soil) | MBC:MBN (atomic ratio) | Gas C (mg C g⁻¹ dry soil) | Approx. CUE |

| | | | | | | | | |
|---|---|---|---|---|---|---|---|---|
| Control | 0.69 | 0.10 | 8.1 | 8.5 | 0.97 | 10.2 | 4.0 | 0.49 |
| N addition | 0.69 | 0.10 | 8.1 | 6.8 | 0.79 | 10.0 | 4.5 | 0.26 |
| P addition | 0.69 | 0.10 | 8.1 | 7.7 | 0.86 | 10.4 | 3.5 | 0.50 |
| NP addition | 0.69 | 0.10 | 8.1 | 9.8 | 1.2 | 9.7 | 3.2 | 0.65 |

### 3.3 Stoichiometric changes in soil solution

We conducted the incubations in the dark without adding any organic carbon substrates. Thus, we interpret the observed changes in the porewater pools of C, N and P to be the result of the processing of existing C, N and P soil pools coupled to the net mineralization of soil organic matter by the resident microbial community. The changes in the pore water DOC concentration and the stoichiometric ratios of DOC to dissolved N and P, respectively, are shown in Fig. 4. The initial microbial biomass stoichiometry is added as a dotted line in each panel of Fig. 4. Overall, the DOC decreased from the initial concentrations, but the remnants were more than the estimated consumption from the cumulative carbon gas emissions, indicating production of DOC from particulate, either through microbial enzymatic decomposition or chemical leaching. The final concentration of DOC in the bog with NP addition ($78.2 \pm 6.88$ mg C $L^{-1}$, mean $\pm$ SD of duplicates) was smaller than the control ($101.6 \pm 9.97$ mg C $L^{-1}$), while the N or P addition was not meaningfully smaller than the control ($97.0 \pm 0.69$ mg C $L^{-1}$ for the N addition; $94.3 \pm 1.19$ mg C $L^{-1}$ for the P addition) (Fig. 4a). In the fen, the final DOC concentrations were all around the control result (Fig. 4b).

The TDN by the end of the incubation was around approximately 40-60% of $NH_4^+$ in both peat types (e.g., $NH_4^+$ 2.3-3.8 mg-N $L^{-1}$ and TDN from 5.2-6.1 mg-N $L^{-1}$ in bog control to N added; $NH_4^+$ 5.2-7.7 mg-N $L^{-1}$ and TDN 13.3-15 mg-N $L^{-1}$ in fen control to N added). In theory, the $NH_4^+$ would have been taken up by plants under field conditions. The majority of the remaining TDN was dissolved organic N with minor contributions by $NO_3^-$ or $NO_2^-$ to the TDN. These porewater patterns may provide some explanation for the reported water quality changes in burned catchments, lasting high phosphate and high dissolved organic N concentrations (Ackley et al., 2021; Burd et al., 2020; Wright et al., 2022).

The stoichiometric ratios of DOC to TDN and TDP in the porewater approached the initial microbial biomass ratios by the end of the incubation of both bog and fen peat. In the bog peat, as the relative N shortage by the initial P addition reduced both MBC growth and cumulative respiration, the CUE was estimated close to the control while the N addition resulted in a much smaller CUE (Table 3). This could be due to the microbial community of bog needing relatively less energy for the necessary N acquisition than for the P acquisition from the peat environment. The bog peat microbial community may have specifically adapted to the constant shortage of available P in peat, allowing them to release as much bioavailable P as required to overcome the nutrient imbalance for C cycling, even with a high energy cost (Hoyos-Santillan et al., 2018). In the fen peat, P release

seemed to be an efficient process for the fen adapted microbial community, given the prompt enzymatic DRP release to meet the increased need by the N addition.

 The largest net microbial biomass growth was observed with the NP addition in the bog peat, presumably because of enhanced assimilation of DOC instead of auxiliary respiration (Giesler et al., 2011; Manzoni et al., 2012; Sinsabaugh et al., 2013; Lin et al., 2014). This is consistent with the reduced $CO_2$ production observed following the NP addition (Fig. 3), despite the larger drop in DOC relative to the N and P additions (Fig. 4). Overall, this implies that NP addition promoted the degradation of soil organic carbon that, under the anaerobic conditions in the soil environment, resulted in the higher cumulative $CH_4$ production.

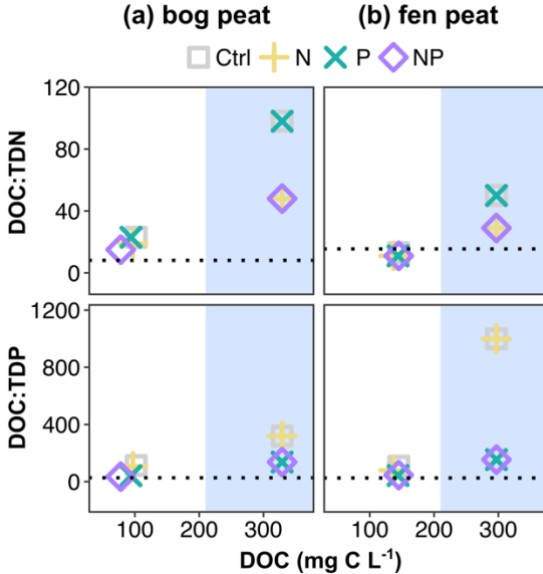

**Figure 4: Porewater dissolved organic carbon and stoichiometric ratio change before and after the incubation. The blue shade indicates where the initial (i.e., at the start of the incubation experiment) compositions are marked in comparison to the compositions at the end of the incubation. The dotted lines represent the initial microbial biomass atomic ratios, measured by microbial biomass carbon (MBC), microbial biomass nitrogen (MBN), and microbial biomass phosphorus (MBP) contents and correspond to bog peat MBC/MBN = 8.1, MBC/MBP = 26.9 and fen peat MBC/MBN = 15.5, MBC/MBP = 25.6 before the incubation.**

Despite the aerobic headspace setting of incubation jars, the water saturated peat environments initiated anaerobic microbial metabolism, as indicated by the net $CH_4$ production and release (Fig. 2). The increase of porewater dissolved iron (Fe) and

355 manganese (Mn) is also indicative of the reductive dissolution of sediment iron and manganese oxides (Table 4). As the DRP concentrations increased in the fen samples, we interpret that some of this DRP production may have been associated with the release of DRP that was bound to Fe and Mn oxides (Table 4). To attribute the net DRP production to Fe and Mn oxide reduction versus biological sources of DRP production such as enzymatic release, we subtracted the amount of DRP release that was possible from Fe and Mn oxide reduction using the maximum molar ratio of 1 mol Fe and Mn to 1 mol DRP (Orihel

et al., 2017). As a result, the largest biological DRP release was measured in the N addition and the smallest amount was released in the NP addition. In the bog peat, net DRP uptake was estimated (Table 4), implying the inherent limitation of available P for microbial communities.

Table 4: Porewater phosphorus changes from the initial (before the incubation) to the final (after the inubation) analysis.

| Porewater | | Initial | | Final | | Final – Initial | | | | Approx. DRP production[a] |
|---|---|---|---|---|---|---|---|---|---|---|
| | | DRP ($\mu$mol L$^{-1}$) | TDP ($\mu$mol L$^{-1}$) | DRP ($\mu$mol L$^{-1}$) | TDP ($\mu$mol L$^{-1}$) | $\Delta$DRP ($\mu$mol L$^{-1}$) | $\Delta$TDP ($\mu$mol L$^{-1}$) | $\Delta$Fe ($\mu$mol L$^{-1}$) | $\Delta$Mn ($\mu$mol L$^{-1}$) | ($\mu$mol L$^{-1}$) |
| Bog | Ctrl | 67.9 | 86.0 | 64.8 | 75.2 | -3.1 | -10.8 | 1.05 | 0.43 | Net uptake |
| | N | 67.9 | 86.0 | 56.9 | 75.6 | -11.0 | -10.4 | 0.89 | 0.41 | Net uptake |
| | P | 182 | 200 | 146 | 193 | -36.0 | -7.0 | 0.79 | 0.46 | Net uptake |
| | NP | 182 | 200 | 160 | 187 | -22.0 | -13.0 | 0.80 | 0.44 | Net uptake |
| Fen | Ctrl | 22.9 | 24.8 | 98.2 | 112 | 75.3 | 87.2 | 15.5 | 6.54 | 53.3 |
| | N | 22.9 | 24.8 | 135 | 146 | 112 | 121.2 | 12.6 | 6.85 | 92.7 |
| | P | 159 | 161 | 217 | 282 | 58.0 | 121.0 | 13.8 | 6.11 | 38.1 |
| | NP | 159 | 161 | 205 | 261 | 46.0 | 100.0 | 14.0 | 5.33 | 26.7 |

[a] **Hypothetical dissolved reactive phosphorus (DRP) production that could be attributed to enzymatic release; for the bog porewater measurements, net DRP uptake was estimated, and thus the DRP production was not determined.**

These observations support the critical role of microbial metabolism in regulating nutrient forms and concentrations in peat porewater and imply the role of peatlands in large scale on the watershed nutrient fluxes and water quality (Liu et al., 2023).

Relying on the internal supply of C substrates from peat decomposition, the fen porewater ended up accumulating additional dissolved inorganic P, possibly in the process of balancing for the bioavailable substrate stoichiometry by microbial activities. In contrast, the net uptake of P by the bog peat in all four of the treatments suggests that bog peatlands might have some capacity to sequester P inputs via microbial assimilation because P is the inherently limiting nutrient for microbial growth (Worrall et al., 2016). However, the stability and time scale of this P sequestration into microbial biomass in the bog soil

remains uncertain.

### 3.4 Limitations

The experimental results reported herein should be considered in the light of some limitations, mainly derived from the small sample size. First, the number of replicates for this type of experiment is recommended three or more, rather than two in this experiment, but this was limited by the amount of soil samples from these remote peatland sites. To test the hypothesis, four

different treatments with subsampling of multiple aliquots before and at the end of the incubation were also needed to measure key soil property changes. Despite sample-size limitations, results from the duplicate samples were consistent in the direction of changes, within instrumental error ranges, and thus the results can be used to infer and discuss environmental implications. Also, in this article, experimental procedures for potential future replication studies are described in detail, so that additional samples and replicates can be tested to assess the generality of the current findings. Second, there was a methodological artefact

in estimating the microbial biomass of the post incubation samples from one of the two study sites. It may have been possible to mitigate this issue by measuring additional aliquots of the incubated samples if prepared for the chloroform treatment at the time. Although this artefact was considered insignificant for the main discussion of this study results, it is encouraged to improve the experimental procedure, adapted in particular for peat soil samples, to avoid this issue reoccurring even for small sample experiments if necessary.

## 4 Conclusions

This study demonstrated that the addition of dissolved N and P to short-term laboratory soil incubations causes changes in microbial C, N, and P cycling with marked differences between fen and bog peat soils. The added availability of dissolved N and P changes the temperature sensitivity ($Q_{10}$) of the soil carbon gas production rates, with an overall decrease in apparent $Q_{10}$ values, which we attribute to a compensatory effect of microbial activity under low-temperature conditions. Given the vast amount of organic-rich peat deposits in northern permafrost regions, where the landscape is rapidly thawing and increasingly experiencing wildfires, scaling up potential perturbations of increased nutrient N and P inputs and changes in nutrient ratios, as well as the long-term consequences for peatland-atmosphere carbon exchanges, will require further research.

The experiment in this study tested the instantaneous responses of peat microbial community to a one-time nutrient nitrogen and phosphorus addition. In a real-world situation, multiple possible scenarios could occur in terms of the variability in the concentrations, ratios, timing, and duration of such nutrients that may be delivered to peatlands, while the combined effects of permafrost thaws and frequent wildfires are likely to increase dissolved nutrients in subarctic peatlands, previously considered remote and less affected by industrial inputs. Future experiments should more systematically investigate various nutrient supply scenarios across different time scales and peatland types for conducting large-scale studies. Understanding the microbial regulation of porewater stoichiometry in response to increased nitrogen or phosphorus availability may be crucial in explaining sudden changes in $CO_2$ and $CH_4$ emissions from peatlands. We recommend future work to explore the existing inherent nutrient limitation of the peatland microbial communities and their response to both one-time and continuous nutrient inputs with a wide range of representative seasonal temperature fluctuations. Consequently, peatland carbon cycle models should also account for the alteration of the temperature sensitivity parameter by shifts in nutrient concentrations and ratios.

## Data availability

The data that support the findings of this study are available at https://doi.org/10.20383/102.0712 in Federated Research Data Repository (FRDR).

## Author contribution

**Eunji Byun**: Conceptualization, Methodology, Investigation, Writing - Original draft preparation. **Fereidoun Rezanezhad**: Conceptualization, Supervision, Writing - Review & Editing. **Stephanie Slowinski**: Conceptualization, Methodology, Validation, Writing - Review & Editing. **Christina Lam**: Methodology, Writing - Review & Editing. **Saraswati Saraswati**: Validation, Writing - Review & Editing, **Stephanie Wright**: Resources, Validation, Writing - Review & Editing. **William L. Quinton**: Resources, Validation, Writing - Review & Editing. **Kara Webster**: Writing - Review & Editing. **Philippe Van Cappellen**: Conceptualization, Supervision, Writing - Review & Editing.

## Competing interests

The authors declare that they have no conflict of interest.

## Acknowledgements

Funding was provided by the Canada Excellence Research Chair (CERC) in Ecohydrology, the Advancing Climate Change Science in Canada program (ACCPJ 536050-18), the Winter Soil Processes in Transition project within the Global Water Futures (GWF) program funded by the Canada First Research Excellence Fund (CFREF), and a Natural Sciences and Engineering Research Council (NSERC) Discovery Grants (RGPIN-2015-03801and RGPIN-2022-03334) and the Can-Peat: Canada's peatlands as nature-based climate solutions project (https://uwaterloo.ca/can-peat) undertaken with the financial support of the Government of Canada. Partial financial support for the first author was provided by the Yonsei University Research Fund (2023-22-0433). The authors wish to acknowledge the support of ArcticNet through the Dehcho Collaborative on Permafrost.

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
