# Peer review of "Effects of nitrogen and phosphorus amendments on CO2 and CH4 production in peat soils of Scotty Creek, Northwest Territories: Potential considerations for wildfire and permafrost thaw impacts on peatland carbon exchanges"

_EGUsphere, 2024_

## Referee Comment (RC2)

The aim of the study is to demonstrate using laboratory incubations how additions of nutrients (N and P salts) affect CO2 and CH4 production in subarctic bog and fen peat. This would indicate the effects of wildfire induced increase in nutrient content in peat. To get more information on the effects of fire on soil processes is relevant because wildfires will increase in northern latitudes with global warming and the control of wildfires in remote areas is difficult.

There are aspects in the text, methods and information given which should be considered when evaluating the output of the study. Enclosed comments.

Title: Title directly suggests that the effects of increasing wildfire and permafrost thaw were studied. The study was conducted with non-fire affected peat samples (?) and merely shows effects and fate of added nutrients in peat. Of course you can discuss the possible links to the wildfire. The title could thus be modified to avoid misleading.

Methods:

Some more information about the sampling sites and peat cores taken should be given. Now there is no data if the cores taken had vegetation cover, or were just bare surfaces sampled?

Line 91: The text indicate that the initially frozen peat cores were placed after thawing in plastic containers and mixed with hands. There are two aspects to be considered, points which can have impact on the results obtained.

The layers of the 0-25 cm peat profile were mixed, and possibly with vegetation? The plant material are serving easily decomposable material for microbes and could enhance their activities and differently in bog peat (moss dominated?) and fen peat (sedge vegetation?). Short-term effects on peat microbiology were studied here without impact of vegetation. If the response of the overall ecosystem is the topic then the measurements should be conducted with intact peat cores including vegetation and primary production. Response of vegetation to nutrients surely has effects also on the microbial activities. The lack of plant activity (carbon release and nutrient uptake) is causing inaccuracies when the aim of the incubations is to mimic non-growing and growing seasons. Or can we assume that in this transition phase plant activity is minor and does not have a great importance? Then it has to be stressed in the aims, even in the title, that the transition phase with minor plant growth is studied here.

The second comments on the mixing of 0-25 cm peat cores is the possibility that layers of different oxygen status in situ were mixed. Then the anaerobes in deeper profile had not their optimum growth conditions in the incubations. Also the populations of aerobic microbes, including methane oxidizers, could have been "diluted" and their real activity was not included to the net release of methane. The water table of the sites should be given. Water status of the peat incubated (lines 110-111) is not clear. Does 80 and 100 % mean water content related to the water holding capacity of the peat?

A basic question for the conclusions is the number of replicates. The two replicates do not allow prober statistical analyses to compare the treatments.

Other comments:

The effects of N and P salts on CO2 and CH4 evolution from two different peat were studied. The measured CO2 evolution reflects CO2 production from anaerobic and aerobic microbial processes. As pointed out above CH4 evolution is the sum of CH4 production and consumption. The peat studied here is taken from subarctic peatlands. There the effects of added nutrients on CO2 and CH4 evolution have not been intensively studied. However, we can well assume that the results on CO2 and CH4 from boreal peatlands treated with nutrients are useful when discussing the results obtained here - the basic mechanisms are the same. However, the literature from boreal peatlands has not been considered here. Especially the effects of N on CO2 and CH4 dynamics have been studied intensively.

A point which could be considered is the effects of salts as such, without any nutrient effects. There are results showing that extra salts can decrease microbial activities in acidic soils. Could this have an impact in the rather short-term incubation experiments with naturally nutrient poor soils? Please note that with fen peat the fumigation extraction method gave negative microbial biomass at the end of the incubation. This means that the amount of extractable organic C was higher before the fumigation than after the fumigation (end of page 10). Would extra salts have been destroyed a substantial part of the microbes releasing their carbon in incubated salt-treated fen peat? Some references if exist could be given about the experiences to apply fumigation extraction method for peat. Are there reports on problems, e.g. negative biomass? If the negative biomass for fen after the incubation is not a result of biomass decomposition, then we can ask if the method has inaccuracies to determine microbial biomass in peat in general (also for bog here)?

In the Fig. 4 microbial biomass carbon and nitrogen are used in the calculations also for fen peat although there is a comment at the end of page 10 that after the incubation the microbial biomass could not be determined for fen with the fumigation extraction method? Did the method give positive biomass for fen before the incubation (initial biomass) but negative biomass just after the incubation (see the comments above for the possible salt effect and methodological problems).

---

## Author Comment (AC1)

**RC1: 'Comment on egusphere-2024-1047', Anonymous Referee #1, 19 Jul 2024**

The authors investigate the effects of nutrient enrichment that comes with wildfires and permafrost thaw in subarctic peat lands on soil CO2 and CH4 production. While they name "subarctic peat lands" in the title, they investigate basically samples from two sites in the same region of Canada. That means the title promises more than is presented in the manuscript. The choice and combination of methods is valid, however, the core concern regarding this manuscript is the fact that the authors used only four samples. I doubt that the informative value of this set-up is sufficient. While the experiment itself is interesting and promising, I strongly recommend to add more samples to the experiment and do/repeat the experiment at least in duplicates, better in triplicates (of every single sample) to check if the obtained data is valid. In the given form I can not recommend the manuscript for publication as I am not convinced that the extent of the experiment is sufficient to support the authors' conclusions. However, it might be an option to address this clearly and adjust the discussion accordingly.

We thank the Reviewer for thoroughly reviewing the preprint and providing constructive criticisms. We would like to confirm that for each of the two peat soils, the incubations were carried out on duplicates [preprint l. 119]. All the raw data have been deposited in the public repository (DOI is given in the Data Availability Statement).

**Completed.** We agree with the reviewer's specific remark on the title of our manuscript. Another reviewer also noted that the title was too broad. We have therefore revised the title to more precisely reflect the scope of our study. It now reads:

"*Effects of nitrogen and phosphorus amendments on CO$_2$ and CH$_4$ production rates in peat soils of Scotty Creek, Northwest Territories, Canada: exploratory incubation results highlight a potential impact of wildfires and permafrost thaw on peatland carbon exchanges*"

We definitely intend to conduct future experiments to assess the generality of the results presented here. Nonetheless, we believe our results, even if preliminary, are timely to communicate an emerging topic with the (peat) soil science community and discuss the need for future research [preprint l. 26-27]. Sharing these preliminary findings through the interactive peer review process of SOIL journal offers one way to achieve such a goal. Thus, we revised the results and discussion section in a concise manner and turned the manuscript in a *Short Communication*. The latter specifically aims to "report new methods or small data sets that have significant implications." (https://www.soil-journal.net/about/manuscript_types.html). In revising our preprint manuscript, we have addressed the reviewer's comments and present the changes here, along with line numbers.

The quality of the language is good and causes at no point problems regarding the comprehensibility. However, the manuscript needs some polishing it should be taken care that a more scientific language is used. The introduction needs to be restructured and rephrased in some parts. Very often, single sentences or parts of the introduction are not connected to each other and therefore the authors' reasoning is not clear. The authors also use very general terms frequently without explaining their meaning in the given context. With the Methods section, it is the same - some parts are more evocative of a staccato than of a coherent text.

**Completed.** Thank you for pointing out the need to improve the writing style and clarity of the manuscript. In the revised version of the manuscript, we made consistent efforts to improve the structure of the Introduction and the explanations in the Methods sections. Also, we have carefully followed the referee's suggestions in the below point-by-point responses.

Please find my detailed comments below:

Title

Having read your manuscript, it is clear what you want to express using this title. However, I suggest to change it slightly to arouse interest in more readers, for instance, "complexity" is a very broad term. In addition, just mentioning "subarctic peatland" suggest that you investigated samples from several subarctic areas - you should clarify this.

**Completed.** We agree and have revised the title accordingly to: "*Effects of nitrogen and phosphorus amendments on $CO_2$ and $CH_4$ production rates in peat soils of Scotty Creek, Northwest Territories, Canada: exploratory incubation results highlight a potential impact of wildfires and permafrost thaw on peatland carbon exchanges*"

Abstract

l. 15-17 Why is an increasing N and P concentration a growing concern in this context? This should be made clear.

**Completed.** The adverse impacts of excessive soil nutrients have been studied in other regions [preprint l.14-15]. We propose that subarctic peatlands may also be experiencing rising exposure to nutrient inputs generated by wildfires and permafrost thawing [preprint l. 15-16]. Due to climate change, wildfires are causing the release of more of the macronutrients nitrogen and phosphorus while permafrost thaw increases the hydrological connectivity within peatland regions [preprint l. 16-17]. However, as the reviewer's comment suggests, the message was not clearly coming through in the preprint article. We have completely revised the abstract, which now reads [revised 31-49]:

"Impacts of nutrient enrichment on soil carbon cycling have been extensively studied in temperate and tropical regions where intensive agriculture and land development have led to large increases in anthropogenic emissions of nitrogen (N) and phosphorus (P). However, how soil carbon sequestration and soil-atmosphere gas exchanges in cold regions respond to greater inputs of N and P remains poorly known, despite recent observations showing significant increases in porewater N and P in burned subarctic peatlands and downstream waters. Wildfires plus enhanced hydrological connectivity due to permafrost thaw have therefore the potential to change carbon turnover and gas emissions in soils of northern peatlands. To start exploring the sensitivity of peatland soil biogeochemistry to variations in N and P availability, we measured the carbon dioxide ($CO_2$) and methane ($CH_4$) production rates during a month-long incubation experiment with soils from a bog and fen collected at the long-term Scotty Creek research station in the Northwest Territories, Canada. Subsamples of the peatland soils were divided into a series of containers to which artificial porewater solutions were added. These solutions were amended with

either dissolved inorganic N, dissolved inorganic P, or both N and P together. Unamended controls were run in parallel. The containers cycled through pre-set temperature steps of 1, 5, 15, and 25°C. Overall, the fen soil yielded higher $CO_2$ and $CH_4$ production rates than the bog soil. Amendment of N to the bog soil produced more $CO_2$ compared to its control, while amendment of P increased $CO_2$ production in the fen soil. Amendment of N and P together reduced $CO_2$ production but increased that of $CH_4$ in both the fen and bog soil incubations. Porewater chemistry at the end of the 30-day experiment showed aqueous C, N, and P stoichiometric ratios that trended toward those of the soil microbial biomasses, hence, implying that the initial microbial nutrient status played a crucial role in determining the responses to the different nutrient amendments. Our preliminary results show that the effects of nutrient enrichment on peatland soil biogeochemistry requires further investigation."

l. 18 I suggest to add "which" here: "...from subarctic bogs and fens, which aimed at..."

**Completed.** We agree with this suggestion and now explicitly mention "subarctic peatlands" [revised. 35] and clarify the provenance of the soils ("Scotty Creek research station in the Northwest Territories, Canada") in the revised abstract [revised l. 39-40].

l. 23-25 Please rephrase this sentence to make it clearer ("It was unexpected..."). Do you mean N and P were added together or both treatments?

**Completed.** Thank you for this comment. We meant the addition of N and P together. We have change the phrasing in the revised manuscript accordingly as follows: "Amendment of N and P together reduced …" [revised l. 44-45].

l. 20-26 It is not really clear how you treated the samples. You had three treatments + control (N, P, N+P, none), but what about the different temperatures you name in line 21? And how long did your experiment last? In line 25 you mention "after a month" - was this the end of the experiment or just the point where you observed that C, N, and P ratios approached initial soil microbial biomass ratios? This needs to be more precise.

**Completed.** We agree that a more precise formulation would be helpful and have revised the abstract accordingly. We invite the reviewer to read the revised abstract above. The full duration of the experiment is now explicitly given ("a month-long incubation experiment"; "at the end of the 30-day experiment"). The temperature cycling through pre-set temperatures is also explicitly stated ("pre-set temperature steps of 1, 5, 15, and 25°C"), as well as the timing of the pore water chemistry ("Porewater chemistry at the end of the 30-day experiment"). [revised l. 39; revised l. 43-44; revised l. 46]

l. 26 What do you mean by "nutrient recycling" here and how do the approaching ratios suggest this? Please explain.

**Completed.** In the revised abstract, we are no longer using the term "nutrient recycling" but point to the observed convergence of the pore water nutrient ratios to those of the microbial biomasses as an indicator of the role of the initial microbial nutrient status as a determinant of the observed responses to the N and P amendments. A full (and admittedly speculative at this stage) explanation

of this observation can be found in the main text of our manuscript (this would be too long to include in the abstract). The statement in the abstract now reads:

"Porewater chemistry at the end of the 30-day experiment showed aqueous C, N, and P stoichiometric ratios that trended toward those of the soil microbial biomasses, hence, implying that the initial microbial nutrient status played a crucial role in determining the responses to the different nutrient amendments." [revised l. 46-49]

Introduction

l. 30-32 In line 30 you name "subarctic regions", in line 31 you refer to "the region". In line 32 you name additionally western Canada, Siberia, and Alaska. Which region is meant here? Do western Canada, Siberia, Alaska not belong to (or contain) subarctic regions? Please check these sentences and clarify your statements.

**Completed.** We agree that a more precise use of geographical terminology is called for. Western Canada, Siberia and Alaska host parts of the subarctic regions. We have now replaced "the region" to "the subarctic regions" (plural) in the revised manuscript to be consistent and explicit [revised l. 53].

l. 36-40 Again, please explain why increasing nutrient inputs are a rising concern here. Above, you only mentioned the increase in POC and DOC, you do not give a reason for the enrichment in N and P.

**Completed.** Upon reflection, we have dropped the "rising concern". Our point is that recent observations have reported substantial enrichment of dissolved nitrogen (N) and phosphorus (P) in western Canada [preprint l. 37-40 and Table 1]. We realize that the statement "as summarized in Table 1" was out of context and not meant to refer to the case studies on DOC and POC increases cases. Thus, we have now corrected the two related sentences in the revised manuscript [the phrase was removed from revised l. 56-57 and placed at revised l. 63 where it is relevant].

Additionally, we have reformulated the motivation for our experiment as follows: "Increasing N and P inputs to peatland soils have the potential to perturb soil carbon cycling in peat soils. For instance, by relieving nutrient limitations on the soil microbial community, additional N and P may accelerate the decomposition of soil organic matter." [revised l. 59-61]

l. 40-44 I suggest to separate the two parts of this sentence and emphasize the effects of permafrost thaw you name here. And still, you do not mention why all of this is a problem. What is the effect of dissolved nutrients? What is the effect of an increased hydrological and geochemical connectivity?

**Completed.** Thank you for this suggestion, which is helpful in clarifying this part. We have revised the text and separated the sentences as follows:

"Thawing permafrost may release additional dissolved nutrients previously bound in frozen sediments (Treat et al., 2019; Schuur et al., 2022; Wright et al., 2022). Furthermore, at the

landscape level, permafrost thaw-induced ground subsidence increases the hydrological and geochemical connectivity among landscape components, including forested areas, fens, thermokarst bogs, and adjacent peatlands (Connon et al., 2014; Gibson et al., 2018; Haynes et al., 2018; Post et al., 2019; Carpino et al., 2021). Thus, there is growing probability for organic carbon stored in peat soils to experience enhanced microbial decomposition driven by nutrient enrichment." [revised l. 64-70]

l. 52 Where does "generally most poor in soil nutrients" refer to? To the N-fixing microbes? And what do you mean by "most poor"?

**Completed.** We admit that 'most poor' did not fully reflect the intended context. We meant to emphasize the nutrient poor status of isolated bogs among the various peatland types in the subarctic and boreal regions. To avoid confusion, especially for those who are not familiar with this ecosystem type (*i.e.,* peatland), we have changed the sentence in the revised manuscript to:

"For example, isolated ombrotrophic bogs rely on direct atmospheric deposition of N and P plus microbial N fixation. Comparatively, minerotrophic channel fens capture nutrients from surrounding water bodies through groundwater and surface water pathways." [revised l. 77-80]

l. 53-56 Here it could help to add one sentence that describes the impacts in temperate and tropical peatland soils.

**Completed.** Thank you for this comment and suggestion. We have now expanded this part in the revised manuscript to describe the impacts of increasing nutrient in temperate and tropical peatland soils with more specific cases:

"Rapid increases in nutrient inputs to peatlands have long been studied in temperate and tropical regions, where there is a higher chance of fertilizer spillover and growing agriculture or fossil fuel-driven atmospheric deposition (Amador and Jones, 1993; Hoyos-Santillan et al., 2018; Lin et al., 2014; Moore et al., 2019; Qualls and Richardson, 2000; Schillereff et al., 2021). Early research on subtropical peatlands in the Everglades, Florida, showed that experimentally P-enriched soils released significantly more $CO_2$ than controls, seemingly because the added nutrient P stimulated soil heterotrophic respiration (Amador & Jones, 1993; Qualls & Richardson, 2000). Recently, however, a study suggested that nutrient availability is not always a rate-determining factor as microbial communities in tropical peatland soils have developed strategies to process site-specific plant litter even without adequate nutrient supply (Hoyos-Santillan et al., 2018). Still, other studies support the important control of nutrient increases in peat decomposition rates. For example, a data synthesis across temperate regional bogs revealed that a long-term P inputs to surface peat enhanced microbial decomposition activities and reduced net organic C burial to deeper peat layers (Schillereff et al., 2021). Another study of a 12-year P fertilization field experiment of temperate mountainous peatlands showed that the extra P supply increased soil respiration and $CO_2$ release, reducing the overall carbon sink function of these peatland ecosystems (Lu et al., 2022)." [revised l. 83-96]

l. 60-63 The connection between the first part ("Recent observations...") and the second one ("but the effects...") is not clear.

**Completed.** Thank you for this comment. We have now revised the sentences to make the connection clear in the revised manuscript by changing the sentences to: "Recent observations highlighted the increase of wildfire impacts, the acceleration of permafrost thaw and the degradation of permafrost stability. As a result, peatland ecosystems and their associated hydrological processes and pathways are rapidly transforming (Gibson et al., 2018). However, the potential effects of sudden nutrient inputs to peatland soils have not been explored for subarctic regions, despite the possible increase of organic carbon decomposition and carbon greenhouse gas emissions suggested by studies in other climate zones." [revised l. 110-115]

l. 64 To me it is still not clear what you mean by "nutrient recycling".

**Completed.** We appreciate this comment on the term "nutrient recycling" used in this manuscript. We meant to describe the microbial utilization of nutrients from in-soil stores (*i.e.,* soil minerals and organic materials, including their necromass), rather than from external inputs. Based on this comment, as well as other detailed comments by the Reviewer, we acknowledge that such recycling cannot be readily inferred from the experimental results presented here. Thus, we have revised the sentence to not include this terminology. It now reads: "In this study, our aim was to investigate the potential acceleration of organic matter decomposition and carbon gas production in subarctic peatland soils following the addition of readily bioavailable dissolved inorganic N and P." [revised l. 115-118].

Methods

Figure 1 Please check the caption. It could also help to mark your sampling area in the small map in the upper left corner.

**Completed.** Thank you for pointing this out. We have revised the caption as well as Figure 1 accordingly. The sampling area is marked in the small inset map in the upper left corner in the revised Figure 1.

[Figure]

**Figure 1:** Scotty Creek watershed bog and fen peat sampling site locations (Map data: Google ©2023 CNES / Airbus, Maxar Technologies) and photos (credit: Mason Dominico). The location and extent of the Taiga Plains Ecozone of Canada (National Ecological Framework for Canada, 2017) is shown in the inset map with a small box indicator for the site locations.

l. 93-96 Please split this sentence in at least two sentences. And please clarify its second part: it should be made clear that the named steps belong to the MgNO3 digestion method.

**Completed.** Thank you for this comment and suggestion. We have now clarified this part in the revised manuscript as follows: "In addition, a sub-sample of each of the peat soils was freeze-dried for analysis of the total organic carbon (TOC), total nitrogen (TN) and total phosphorus (TP) concentrations. TOC and TN were determined using a CHNS analyzer (Carlo Erba NA-1500 Elemental Analyzer; detection limit of 1% by mass for both TOC and TN). TP was determined following the method of Aspila et al. (1976): an aliquot of soil was ashed in a muffle furnace at 500 °C with magnesium nitrate added as an oxidant. The ashed soil was subsequently extracted by mixing with 1 M hydrochloric acid on a shaker for 16 hours. The extract was then analyzed for total dissolved P by ICP-OES (Thermo Scientific iCAP 6300 Duo)." [revised l. 148-155].

l. 96-97 I do not understand the last sentence here. Do you mean you did not attempt to compare the pools prior to the treatment with the pools after the treatment?

**Completed.** Thank you for this comment. The sentence meant to express technical concerns with expressing the TOC, TN and TP mass concentrations relative to the peat soil weight, given the contributions to the total soil weight from salts. However, this is considered to have had very minor effects on the results presented. In addition, all measured values are accessible in the associated public dataset. Given the potential confusion our original comment may create, we have deleted this sentence in the revised manuscript. [revised l. 155-156]

l. 101-104 Please rephrase this part.

**Completed.** Thanks for this suggestion. We have now revised this part in the revised manuscript as follows: "The pre-incubation caused some reduction of the peat moisture contents. We therefore re-measured the moisture contents at the end of the pre-incubation period by the oven-drying method to determine how much porewater solutions needed to be added to return the peat moisture contents to the selected levels. Note that the latter were selected to be close to average field conditions." [revised l. 160-165]

l. 105-106 Why did you measure microbial biomass C, N, and P after the pre-incubation?

We estimated the microbial biomass C, N, and P contents after the pre-incubation to directly compare the elemental contents at the start and the end of the incubation. This enabled us to see the impact of the various nutrient treatments of the microbial nutrient status (i.e., the end of the pre-incubation period yielded the baseline nutrient status). We have clarified this point in the revised manuscript: "After the pre-incubation, each of the bog and fen peat samples was subsampled for fumigation to measure microbial biomass C, N and P concentrations. These concentrations were considered the baseline values against which changes in microbial C, N, and P accompanying the various nutrient treatments were evaluated." [revised l. 167-169]

l. 102-103/108-111 You repeated the measurement of moisture content after pre-incubation to decide how much porewater solution should be added, but then you use data from adjacent meterorological stations to obtain this information? Please explain what you did clearly and in comprehensible steps.

**Completed.** Thank you for this comment. We have now clarified the steps in the revised manuscript as follows: "The moisture conditions for the incubation experiment were selected using field-based growing-season averages measured in adjacent meteorological stations equipped with soil sensors. The selected moisture conditions were a volumetric water content of 80% for the bog peat and 100% for the fen peat. These values were then used to calculate the volumes of artificial porewater to add to the experimental containers. [revised l. 174-178]

l. 165-166 "The subsamples were treated either with chloroform for 24 hours to fumigate in a vacuum desiccator, or with no fumigation."? Please explain why, this way it sounds like you just randomly fumigated some samples and some not.

**Completed.** Thank you for this comment. We followed the standard fumigation method described in the literature (Brookes et al., 1984; Vance et al., 1987; Joergensen, 1996; Jenkinson et al., 2004 cited in the preprint). We have now clarified the experimental procedure in section 2.3.2 of the revised manuscript. [revised l. 228-244]

Results and Discussion

l. 190 It should be "resulted in" (whole section, not only this line).

**Completed.** Thank you for this comment. We have now made the corrections throughout the revised manuscript.

l. 192-194 "...are consistent with its higher microbial biomass than the bog initially."? Please check again thoroughly your manuscript if references within sentences are correct.

**Completed.** Thank you for this comment. We have now changed this statement to: "The higher $CO_2$ and $CH_4$ production rates in the fen soil are consistent with the higher initial biomass of the fen soil (MBC 4.3 mg C $g^{-1}$) compared to that of the bog soil (MBC 0.7 mg C $g^{-1}$)." [revised l. 266-267]

l. 196-203 It would massively enhance the comprehensibility if you would focus on one treatment after the other.

**Completed.** Thank for this comment and suggestion. We have changed the text accordingly to [revised l. 278-285]: "Temperature variably affected the gas production rates in the different nutrient treatments. The bog peat soil with added P and NP (thus, P amendment effects in general) yielded distinctively lower $CO_2$ production rates at 25°C but higher $CO_2$ production rates at 15°C, compared to the no-nutrient control (Fig. 2a). The fen peat soil with added N and NP (thus, N amendment effects in general) resulted in lower $CO_2$ production rates at 25°C but no apparent difference at 15°C, compared to the no-nutrient control. For the fen peat soil, the P only amendment (i.e., without N) resulted in distinctively higher $CO_2$ rates than the N amended treatments (N only and NP) and the control (Fig. 2b). The $CH_4$ production rates of the fen soil

also showed noticeable temperature effects with the greatest differences between the various nutrient treatments observed at 25°C (Fig. 2b)."

l. 216 Why do you refer to Table 2 here?

**Completed.** Thank you for noting this. This was a mistake and it has been now corrected. The reference to Table 2 was not meant for this particular sentence, but for subsequent ones. [revised l. 299]

Table 2 Given are the Q10 values - why don't you mention this in the caption? Why are there no values for bog CH4?

**Completed.** Thank you for this comment and suggestion. We have now added a sentence to the methods section to explain why no $Q_{10}$ values were given for bog peat $CH_4$ effluxes: "However, the $CH_4$ production rates in the bog peat containers were very low and yielded too little variation to derive meaningful $Q_{10}$ values for $CH_4$ production." [revised l. 295-296]

l. 222 Here, it is 3.89 - in the table it is 3.90. In gnereal: Is your method sufficiently precise to give two decimals?

**Completed.** We appreciate the reviewer's concern. Providing two decimals for the $Q_{10}$ estimations may give a false impression of the representativeness of the reported flux rates considering the limited number of sampling locations and number of experimental incubations. Therefore, we now present the $Q_{10}$ values to one significant digit in the revised manuscript [revised Table 2]. Note that the individual gas flux measurements themselves are very precise and are thus given with the appropriate decimals in the open dataset made available with the manuscript.

Figure 3 Your caption should make clear which differences are given: "Relative differences (%) between x and y in the..."

**Completed.** Thank you for this comment and suggestion. We have now revised the caption to: "Relative differences (%) in cumulative $CO_2$, $CH_4$, and total carbon gas (Gas C) emissions integrated over the entire incubation experiment: comparing the unamended control (Ctrl) and the nutrient amended treatments N (N only), P (P only) and NP (both N and P together)." [revised l. 337-339]

l. 263-270 These are very general statements and should be better connected to your data.

**Completed.** Thank you for this comment. We agree that these general background statements related to possible organic carbon changes by microbial processes are not point-by-point connected with the data of this study. Accordingly, we have decided to move this part of the text to the Introduction where it now appears before stating the specific aims of our study. Subsequently, we also rephrased some parts for the context: "Given the high accumulation of plant organic materials in peatlands, the absolute amount of carbon as substrates for microbial activities is assumed not limiting in peat soils. Microbial strategies to overcome relative nutrient imbalances of N, P or other nutrients relative to carbon may fall into three broad categories: (1) increasing the production of

extracellular enzymes to acquire necessary nutrient elements from enzyme-facilitated organic matter degradation, (2) recycling of the nutrient elements assimilated in dead microbial cells (i.e., necromass recycling), and (3) releasing some extra carbon in relation to the other major nutrients via auxiliary respiration without biomass assimilation (i.e., overflow respiration) (Giesler et al., 2011; Manzoni et al., 2012; Lin et al., 2014; Hoyos-Santillan et al., 2018; Schillereff et al., 2021; Lu et al., 2022). Laboratory soil incubation experiment to measure the changes in carbon gas production rates with net microbial biomass change can help address which strategy is activated. [revised l. 98-106]

l. 273-275 Please rephrase this sentence.

**Completed.** Agreed – we have now rephrased this sentence in the revised manuscript to: "We conducted the incubations in the dark without adding any organic carbon substrates. Thus, we interpret the observed changes in the porewater pools of C, N and P to be the result of the processing of existing C, N and P soil pools coupled to the net mineralization of soil organic matter by the resident microbial community. The changes in the pore water DOC concentration and the stoichiometric ratios of DOC to dissolved N and P, respectively, are shown in Fig. 4." [revised l. 363-368]

Figure 4 "The shades ara indicates where the initial porewater ratios are marked"?

**Completed.** Thank you for this comment. We have now revised the caption of Figure 4 to clarify this as: "The blue shade indicates where the initial (i.e., at the start of the incubation experiment) compositions are marked in comparison to the compositions at the end of the incubation." [revised l. 399-401]

---

## Author Comment (AC2)

**CC1: 'Comment on egusphere-2024-1047', Nasrollah Sepehrnia, 27 Jul 2024**

This study investigates the impact of nutrient enrichment on carbon gas production in subarctic peatlands. Authors have examined addition of nitrogen and phosphorus to soil samples from bogs and fens, and observed varied effects on CO2 and CH4 production, indicating complex interactions between nutrient levels and carbon cycling in these regions. The results have illustrated the importance of nutrient enrichment on microbial activity and carbon turnover in nutrient-poor subarctic soils. I think this lab experiment can provide strong evidence for conducting large-scale studies. I recommend the manuscript for publication after consideration of the following comments (minor revisions).

We thank the community reviewer for reading our preprint and providing suggestions for improvement. We have made the suggested changes in the revised manuscript. Note that this includes the pertinent remark that the lab results presented support conducting large-scale field studies. We have added this as one of our conclusions in section 4 (Future research suggestions) of the revised manuscript. "Future experiments should more systematically investigate various nutrient supply scenarios across different time scales and peatland types for conducting large-scale studies." [revised l. 433-434]

1- Introduction

Authors may provide readers with clear hypotheses (in one or two sentences) and objectives afterward " …..but the effects of sudden nutrient inputs to the peatland soils remain poorly understood….."

We have improved the phrasing of our study goals accordingly, as follows:

"In this study, our aim was to investigate the hypothesis that the addition of dissolved inorganic N and P enhances organic matter decomposition and carbon gas production in subarctic peatland soils. We further anticipated that variations in nutrient ratios (N:P, C:N, C:P) and among different soils, and the associated adaptations of the soil microbial communities would contribute to understanding the short-term responses of different types of peatlands to additional N or P (Hill et al., 2014; Hoyos-Santillan et al., 2018). A laboratory soil incubation experiment was conducted to compare the $CO_2$ and $CH_4$ production rates in the field-sampled bog or fen soils under various treatments: control, N only, P only, and both N and P (NP hereafter) addition. "Fluctuating temperatures were imposed during the month-long incubation to analyze the temperature sensitivity of the gas production rates in the different nutrient treatments." [revised l. 122-124]

-Authors may rephrase lines 60-70 to highlight the following hypotheses accordingly.

**Completed:** Thanks for this comment. We have revised the introduction section according to the specific comments from Referee #1, and believe the changes improved the expression of the hypotheses.

-Why nutrient addition in subarctic peatland soils?

The research context for the experiments with added nutrients (dissolved N and P) is explained in the introduction section of the preprint. In summary, there have been recent observations of increased nutrient levels in the study region that have been attributed to ongoing permafrost thaw. Given the accelerating permafrost thaw in the subarctic regions, more peatland soils are likely to experience nutrient enrichment, which in turn may alter the biogeochemistry of those organic rich soils (especially in regard to soil organic carbon decomposition and carbon gas release).

-Microbial Responses in the short-term and the impacts?

Given the limitations of lab-based incubation experiments, we restricted ourselves to monitoring the potential immediate responses of peatland soils to added N and P. Our short-term exploratory study can provide a basis for future research on long-term impacts of nutrient enrichments, including though in-field manipulation studies.

-Nutrient enrichment and GHG emission in subarctic peatland soils?

The impacts of nutrient enrichment in various soil environments have been studied in relation to direct human-driven pressures such as agricultural expansion and fertilizer use. However, the impacts of nutrient enrichment in remote subarctic peatland soils remain largely unexplored, despite evidence that permafrost thaw and expanding forest fires can increase the delivery of N and P. Thus, with our study, we wish to raise awareness of this issue, especially given the large amounts of organic carbon stored in northern peatlands.

**2- Materials and methods**

This part is organized very well.

Thank you for your positive comment about the Materials and Method section.

**3- Results and discussion**

The results are described very well using the illustrated figures and tables. Authors may compare their results with other relevant studies (e.g., from the cited Refs) in the following parts.

 "3.1 Transient changes in the soil CO2 and CH4 production and, 3.2 Variations in the temperature sensitivity of CO2 and CH4 productions"

**Completed.** Thanks for your comment and suggestion. We have revised the manuscript, and the findings are now discussed in comparison with some recent studies we have cited. The revised text now reads:

 "The temperature sensitivities estimated here suggest that the alleviation of nutrient limitation may result in either relatively high $CO_2$ production rates at the lower temperatures tested (according to the low $Q_{10}$ values for both NP, bog P, and fen N), or high $CO_2$ production rates at the higher-end temperatures (according to the high $Q_{10}$ for bog N and fen P), or some

combination of both trends. Recently, Liu et al. (2022) suggested that the catalytic efficiency of hydrolytic enzymes involved in soil N and P recycling significantly increases with increasing temperature. This would imply that microbial respiration at warmer temperature would become less dependent on the initial soil and N and P stores. However, as shown by our results, the coupled effects of nutrient limitation and temperature may be more complex than currently recognized. A better predictive understanding of the variations in $Q_{10}$ values across peatland classes remains a crucial task for the calibration of process-based C cycle models of peatlands (Bona et al., 2020). For example, the recently observed latitudinal increase of $Q_{10}$ values for $CO_2$ production in peat soils (Byun et al., 2021) could point to an increasing severity of nutrient limitation for peatlands at high latitudes." [revised l. 309-322]

---

## Author Comment (AC3)

**CC2: 'Comment on egusphere-2024-1047', Nasrollah Sepehrnia, 27 Jul 2024**

Please note that the community comment posted here (CC2) is identical to the CC1 above. Please refer to Reply to CC1 for our replies to the specific comments from this reviewer.

---

## Author Comment (AC4)

**RC2: 'Comment on egusphere-2024-1047', Anonymous Referee #2, 24 Sep 2024**

The aim of the study is to demonstrate using laboratory incubations how additions of nutrients (N and P salts) affect CO2 and CH4 production in subarctic bog and fen peat. This would indicate the effects of wildfire induced increase in nutrient content in peat. To get more information on the effects of fire on soil processes is relevant because wildfires will increase in northern latitudes with global warming and the control of wildfires in remote areas is difficult.

There are aspects in the text, methods and information given which should be considered when evaluating the output of the study. Enclosed comments.

We would like to thank the Reviewer for evaluating our preprint manuscript with consideration of the context of our research. We agree with the Reviewer's general and specific comments and suggestions and believe we have addressed all the comments in the revised manuscript. In the following paragraphs, each comment is addressed individually. We considered all the points raised by the Reviewer and have detailed the ways in which we have addressed them in the revised version of the manuscript.

Title: Title directly suggests that the effects of increasing wildfire and permafrost thaw were studied. The study was conducted with non-fire affected peat samples (?) and merely shows effects and fate of added nutrients in peat. Of course you can discuss the possible links to the wildfire. The title could thus be modified to avoid misleading.

**Completed.** We agree with this suggestion and have now revised the title to:

"*Effects of nitrogen and phosphorus amendments on $CO_2$ and $CH_4$ production rates in peat soils of Scotty Creek, Northwest Territories, Canada: exploratory incubation results highlight a potential impact of wildfires and permafrost thaw on peatland carbon exchanges*"

Methods:

Some more information about the sampling sites and peat cores taken should be given. Now there is no data if the cores taken had vegetation cover, or were just bare surfaces sampled?

**Completed.** Thank you for this comment and we have made sure to give complete descriptions of the sampling sites and the soil coring. Both sites were covered by vegetation at the time of sampling as shown in Fig. 1. The corresponding text reads: "In October 2020, two duplicate peat soil cores (0-25 cm) were taken from each of the following two sites: a thermokarst bog covered by Sphagnum mosses (hereafter referred to as 'bog peat' or 'bog soil') and a channel fen covered by herbaceous plants (hereafter referred to as 'fen peat' or 'fen soil') as shown in Fig. 1." [revised l. 131-133]

Line 91: The text indicate that the initially frozen peat cores were placed after thawing in plastic containers and mixed with hands. There are two aspects to be considered, points which can have impact on the results obtained.

The reviewer is correct: the peat samples from each site were manually mixed prior to the incubation experiment. They were thawed and mixed inside the original sampling containers to avoid loss of any in-situ materials, including pore water.

The layers of the 0-25 cm peat profile were mixed, and possibly with vegetation? The plant material are serving easily decomposable material for microbes and could enhance their activities and differently in bog peat (moss dominated?) and fen peat (sedge vegetation?). Short-term effects on peat microbiology were studied here without impact of vegetation. If the response of the overall ecosystem is the topic then the measurements should be conducted with intact peat cores including vegetation and primary production. Response of vegetation to nutrients surely has effects also on the microbial activities. The lack of plant activity (carbon release and nutrient uptake) is causing inaccuracies when the aim of the incubations is to mimic non-growing and growing seasons. Or can we assume that in this transition phase plant activity is minor and does not have a great importance? Then it has to be stressed in the aims, even in the title, that the transition phase with minor plant growth is studied here.

To effectively address the points raised by the reviewer, we have broken our responses into three parts:

**Point 1)** "The layers of the 0-25 cm peat profile were mixed, and possibly with vegetation?"

The soil profiles on peat plateaus at Scotty Creek have broadly two distinct layers, an upper layer containing living and lightly decomposed organic matter, and a lower layer of peat in a more advanced stage of decomposition. Although the distinction of peat layers is not always clear, we removed the topmost surface layer with fresh, large plant debris from each core. Although it may have left some plant debris, it was not our aim to test the activity of live vegetation, which was also not possible by freezing the cores after the field sampling to transport them to our lab in Onatrio. We now state this  explicitly in the revised manuscript. "The top layer containing fresh, large plant debris was removed from each peat core." [revised l. 145-146]

**Point 2)** "The plant material are serving easily decomposable material for microbes and could enhance their activities and differently in bog peat (moss dominated?) and fen peat (sedge vegetation?)."

We agree with the reviewer. In our study, we focussed on the effects of nutrient enrichment on the decomposition of the soil organic matter. Increased decomposition means less carbon preservation and enhanced soil $CO_2$ and $CH_4$ emissions, and vice versa for decreased decomposition. The reviewer is correct that the different vegetation types deliver organic materials of different reactivities to the soil microbial communities and, hence, play a crucial role in soil respiration and fermentation processes.

**Point 3)** Short-term effects on peat microbiology were studied here without impact of vegetation. If the response of the overall ecosystem is the topic then the measurements should be conducted with intact peat cores including vegetation and primary production. Response of vegetation to

nutrients surely has effects also on the microbial activities. The lack of plant activity (carbon release and nutrient uptake) is causing inaccuracies when the aim of the incubations is to mimic non-growing and growing seasons. Or can we assume that in this transition phase plant activity is minor and does not have a great importance? Then it has to be stressed in the aims, even in the title, that the transition phase with minor plant growth is studied here."

As stated above, we would like to emphasize that the aim and scope of this study was not to examine the overall ecosystem response to nutrient amendments. Rather, we want to bring attention to the potential changes increased inputs of N and P may cause on the ability of the resident microbial populations to degrade the soil organic carbon store. Whole ecosystem carbon gas exchanges are better captured my measurements with flux towers or closed chamber systems as reported, for example, by Chasmer et al. (2012) who compared net chamber $CO_2$ fluxes among the fens, bogs and peat plateaus at Scotty Creek (note; we have added this reference to the revised manuscript [revised l. 267-269]). Soil organic matter incubation experiments, such as those presented in our study, provide complementary data that help interpret the ecosystem fluxes.

Chasmer, L., Kenward, A., Quinton, W., & Petrone, R. (2012). $CO_2$ Exchanges within Zones of Rapid Conversion from Permafrost Plateau to Bog and Fen Land Cover Types. Arctic, Antarctic, and Alpine Research, 44(4), 399–411. https://doi.org/10.1657/1938-4246-44.4.399

The second comments on the mixing of 0-25 cm peat cores is the possibility that layers of different oxygen status in situ were mixed. Then the anaerobes in deeper profile had not their optimum growth conditions in the incubations. Also the populations of aerobic microbes, including methane oxidizers, could have been "diluted" and their real activity was not included to the net release of methane. The water table of the sites should be given. Water status of the peat incubated (lines 110-111) is not clear. Does 80 and 100 % mean water content related to the water holding capacity of the peat?

We thank the reviewer for these insightful comments. These and similar ones by the other reviewers have made us thoroughly revise our methods section to remove any ambiguities about the experimental procedures and conditions and their relationships with in-field conditions. Specifically, we now include the following statement:

"The moisture conditions for the incubation experiment were selected using field-based growing-season averages measured in adjacent meteorological stations equipped with soil sensors. The selected moisture conditions were a volumetric water content of 80% for the bog peat and 100% for the fen peat. These values were then used to calculate the volumes of artificial porewater to add to the experimental containers." [revised l. 174-178]

Note that the water contents of the peat samples were measured gravimetrically (i.e., drying of the sample to measure the water mass loss), and the resulting values converted to volumetric ratios (the conversion equation and related data values are provided in the linked public dataset – DOI given in the Data Availability Section). The water contents for the incubation samples were those

corresponding to saturation to reflect the field conditions. This aligns with the goal of the experiment to test nutrient enrichment scenarios relevant to possible conditions created by the combined occurrence of wildfires and increased hydrological flows due to thermokarst intensification. Admittedly, our experimental scenarios are not exhaustive and cannot simply be translated to whole-ecosystem impacts in northern peatlands. Nonetheless, we feel that the real value of our findings is to highlight the need for further studies on nutrient enrichment in these peatlands. This is also the main reason for choosing publication as a *Short Communication* type article in the SOIL journal to reach researchers interested in global soil and environmental changes.

A basic question for the conclusions is the number of replicates. The two replicates do not allow prober statistical analyses to compare the treatments.

We agree and caution against over-generalizing the experimental results. We were limited by the amounts of peat soils available and were only able to run duplicates. Note, however, that each incubation reservoir cycled through several temperature steps. The $CO_2$ and $CH_4$ production rates measured during these individual temperature steps contribute to assessing the internal consistency of the results.

Other comments:

The effects of N and P salts on CO2 and CH4 evolution from two different peat were studied. The measured $CO_2$ evolution reflects CO2 production from anaerobic and aerobic microbial processes. As pointed out above CH4 evolution is the sum of CH4 production and consumption. The peat studied here is taken from subarctic peatlands. There the effects of added nutrients on CO2 and CH4 evolution have not been intensively studied. However, we can well assume that the results on CO2 and CH4 from boreal peatlands treated with nutrients are useful when discussing the results obtained here - the basic mechanisms are the same. However, the literature from boreal peatlands has not been considered here. Especially the effects of N on CO2 and CH4 dynamics have been studied intensively.

We agree that the effects of N deposition in boreal peatlands have been studied in relation to plant-soil interactions. The aim of our study, however, was to compare the impacts of both N and P additions to soil organic matter degradation, within the context of intensifying wildfires and thermokarst (e.g., Gustine et al., 2022 plus the other articles cited in the preprint). The conditions addressed by our study are quite unique for the subarctic regions and more akin to the direct runoff of nutrient enriched waters seen in tropical and temperate regions impacted by direct human activities, rather than by long-range atmospheric deposition that has received most attention in northern high latitudes (e.g., Utstøl-Klein et al., 2015; Gong et al., 2019; Leroy et al., 2019; Gong et al., 2024). Thus, we believe our experimental approach may offer a more relevant starting point to address nutrient enrichment in cold regions that are remote from direct anthropogenic land (but that experience accelerating climate warming and the associated increases in permafrost thaw and wildfires). In the final analysis, we refrained from linking our study too closely with those on

boreal peatland while still including them in the referenced literature (e.g., Utstøl-Klein et al., 2015; Gong et al., 2019; Leroy et al., 2019; Gong et al., 2024).

Gustine, R.N., Hanan, E.J., Robichaud, P.R. et al. (2022), From burned slopes to streams: how wildfire affects nitrogen cycling and retention in forests and fire-prone watersheds. Biogeochemistry 157, 51–68. https://doi.org/10.1007/s10533-021-00861-0

Gong, Y., Wu, J., Vogt, J., Le, T.B. (2019), Warming reduces the increase in N2O emission under nitrogen fertilization in a boreal peatland. Sci Total Environ 664, 72–78. https://doi.org/10.1016/j.scitotenv.2019.02.012

Leroy, F., Gogo, S., Guimbaud, C., et al. (2019), Response of C and N cycles to N fertilization in Sphagnum and Molinia-dominated peat mesocosms. J Environ Sci 77, 264–272. https://doi.org/10.1016/j.jes.2018.08.003

Gong, Y., Wu, J., Roulet, N., Le, T. B., Ye, C., & Zhang, Q. (2024), Vegetation composition regulates the interaction of warming and nitrogen deposition on net carbon dioxide uptake in a boreal peatland. Functional Ecology, 38, 417–428. https://doi.org/10.1111/1365-2435.14480

Utstøl-Klein, S., Halvorsen, R. and Ohlson, M. (2015), Increase in carbon accumulation in a boreal peatland following a period of wetter climate and long-term decrease in nitrogen deposition. New Phytol, 206: 1238-1246. https://doi.org/10.1111/nph.13311

A point which could be considered is the effects of salts as such, without any nutrient effects. There are results showing that extra salts can decrease microbial activities in acidic soils. Could this have an impact in the rather short-term incubation experiments with naturally nutrient poor soils?

The reviewer raises a very good point. Our results do not support a major effect of salinity, however. If the extra salts would have caused a decrease in microbial activity, the N, P and NP amendments (as N and P salts) should have shown decreased carbon gas production rates compared to the no-nutrient (and thus no-salt added) controls, which is not supported by our results. Also, the microbial biomass increase was highest in the bog soil with the highest amount of salt added through both N and P amendments (Table 3).

Please note that with fen peat the fumigation extraction method gave negative microbial biomass at the end of the incubation. This means that the amount of extractable organic C was higher before the fumigation than after the fumigation (end of page 10). Would extra salts have been destroyed a substantial part of the microbes releasing their carbon in incubated salt-treated fen peat?

**Completed.** The reviewer is correct: the negative biomass is based on the sample weight measurements before and after the fumigation. It is hard to imagine that the organic matter decomposition inferred from the $CO_2$ and $CH_4$ production would occur without microbes. Thus, the inferred negative biomass for the fen soil is likely a methodological artefact, as we now state in the revised manuscript: In Methods, "The resulting values from these sample measurements were compiled into a calculation spreadsheet to derive MBC, MBN and MBP values for each peat

sample (Byun et al., 2024)." [revised l. 243-244] and in Results and Discussion, "It is difficult to imagine that the observed $CO_2$ and $CH_4$ production throughout the incubation period could have occurred without microbes in fen soil samples. Thus, the inferred negative biomass for the fen soil was considered a methodological artefact." [revised l. 343-345] (including a reference to the public accessible dataset for this experiment).

Byun, E., Rezanezhad, F., Slowinski, S., Lam, C., Saraswati, S., Wright, S., Quinton, W., Webster, K., Van Cappellen, P.: Dataset for Examining the Effects of Nutrient Pulses on Biogeochemical Cycling in Subarctic Peatlands in the Context of Permafrost Thaw and Wildfires. Federated Research Data Repository. 10.20383/102.0712, 2024

Some references if exist could be given about the experiences to apply fumigation extraction method for peat. Are there reports on problems, e.g. negative biomass? If the negative biomass for fen after the incubation is not a result of biomass decomposition, then we can ask if the method has inaccuracies to determine microbial biomass in peat in general (also for bog here)?

**Completed.** We appreciate this point and see the need of exploring this issue further. A negative biomass result is not entirely surprising given the method's multiple steps of sample treatment and repeated weight measurements each introducing some measure of uncertainty. We have further clarified this aspect of the fumigation method in the revised manuscript as well as in our replies to the above comment [revised l. 243-244; 343-345]. Please also note that we only had limited amounts of soil sample for the fumigation methods (several aliquots were needed for the other chemical analyses).

In the Fig. 4 microbial biomass carbon and nitrogen are used in the calculations also for fen peat although there is a comment at the end of page 10 that after the incubation the microbial biomass could not be determined for fen with the fumigation extraction method? Did the method give positive biomass for fen before the incubation (initial biomass) but negative biomass just after the incubation (see the comments above for the possible salt effect and methodological problems).

**Completed.** The reviewer is correct: the fumigation method gave a positive initial value for the fen soil. Full details for the method and the obtained values are given in the associated open dataset "The resulting values from these sample measurements were compiled into a calculation spreadsheet to derive MBC, MBN and MBP vales for each peat sample (Byun et al., 2024)" [revised l. 243-244]. For this reason (and those above), we used the initial biomass elemental compositions to interpret the compositional changes in the nutrient porewater ratios that occurred over the course of the experiment. We have clarified this in the revised manuscript. "We conducted the incubations in the dark without adding any organic carbon substrates. Thus, we interpret the observed changes in the porewater pools of C, N and P to be the result of the processing of existing C, N and P soil pools coupled to the net mineralization of soil organic matter by the resident microbial community. The changes in the pore water DOC concentration and the stoichiometric ratios of DOC to dissolved N and P, respectively, are shown in Fig. 4. The initial microbial biomass stoichiometry is added as a dotted line in each panel of Fig. 4." [revised l. 363-368]

---

## Referee Report (RR1)

**Review of manuscript "Effects of nitrogen and phosphorus amendments on CO2 and CH4 production rates in peat soils of Scotty Creek, Northwest Territories, Canada: exploratory incubation results highlight a potential impact of wildfires and permafrost thaw on peatland carbon exchanges" by Eunji Byun et al.**

I enjoyed reading the manuscript and I found the study well designed and the manuscript overall written well and definitely worthy consideration for publication. It still, nevertheless, requires some careful revision and (sometimes) restructuring, to better lead the readers through to complex methodology and comprehensive (hence, also confusing) results and especially discussion. I recommend the manuscript for publication, after a further revision, and I encourage the authors to complete the great task they started so well. Please see my more detailed comments below:

**Title: I find is somewhat long, even though I understand it was corrected based on previous comments. Nevertheless, it is strikingly informative, so I suppose it is mostly for the editor and the authors to decide if to leave it so long or maybe cut the last part following the colon.**

**Abstract: overall well written, well summarises the project and presents its aims. The only sentence I would recommend to reconsider is the last one – all our studies are not final, and more research can always help, but I doubt this is the only strong conclusion you can make out of this experiment. I can see that in the previous round of reviews you received some harsh criticism, which I found unmerited. Your findings have an important value and should have a corresponding concluding sentence in the end of the abstract.**

**Introduction: overall good, I only have one comment related to Table 1 and the style of presenting the references**

Line 45. I would contest the need of the Table 1 to prove your point here, and I would recommend moving the table to supplementary materials. While the table shows important examples of nutrient enrichment after wildfires, it is sightly out of scope for your study's objectives, and I doubt you must have it there, in the main text, to justify your experiment. You are doing an excellent justification with your introduction text already. I would rather focus on more clear presentation of the references for the first sentences of this paragraph.

This comment is applicable for the entire text – the reader is often left with lengthy stretches of text without references, followed with a bunch of references in one place (e.g., Emelko et al., 2016; Van Beest et al., 2019; Emmerton et al., 2020; Orlova et al., 2020). In such cases, readers might be left confounded – do all these mentioned studies prove one statement (as it is in line 50; in which case, it is not necessary to show that many references, 1-3 is usually enough) or all 3 to 5 statements before that (as in lines 45, 80, etc... in which case, it would be beneficial to attribute the references to their corresponding statements).

**Methods. It was detailed enough and mostly easy to follow. Adding a graphic summary or a table pointing to each step done before and during the experiment would greatly help readers, but I would not insist on adding one, it is just an idea.**

**Section 2.1.** I recommend adding some essential information that would help readers to make better interpretations of the results. Firstly, you mention that the study site is within discontinuous permafrost area. It would be important to indicate whether the two peatlands you study are affected by permafrost, and most importantly, whether the coring sites are underlain by permafrost. Secondly, you sampled in October, so it would be worth mentioning if the peat samples were already frozen when you sampled, and if they remained frozen during the travel to Waterloo.

**Section 2.2. Line 121.** Was the bulk density used anywhere in the study? Judging from the method you used to estimate it, it might not be the best variable to present. As I did not find any further mentions of density in the manuscript, I would recommend removing it from the methods as well – it slightly reduces the quality of the study, when mentioned like that. But if you used density somewhere else, then keep it.

**Results and Discussion. I would recommend a detailed revision of this part. You have a good study design and diverse data to present. The latter often causes difficulties, when writing a clear discussion, and this was the main issue I identified while reading your manuscript. If it helps, you may consider separating results from discussion; otherwise, please carefully revise each section, and make sure you do not limit yourself to commenting on each particular detail of presented results but also expand into a larger context. In many cases, a presentation of what is normally expected for different variables would make the discussion clearer (not everyone reading the article will be as knowledgeable in all the domains you touch). Overall, I missed a discussion on the meaning of your findings – what did we learn from this, where do we go here? May we expect similar wildfire effects in other subarctic wetlands: what will be the effect of permafrost disappearance, or increasing peatland ground temperatures? Please, do not get discouraged by the previous reviewer's comments about your experiment being unrepresentative. While it is admirable that you carefully acknowledge the limitations of the study, we often must work with what we have, and careful interpretations (based on similar studies) are acceptable. P.S., fens and bogs are highly representative for boreal regions and comparing these two is a significant contribution to the scientific knowledge, even if you work with only one particular study site. Boreal/subarctic regions are strikingly similar across North America and Eurasia.**

I am aware that you have submitted for a short communication, so my preceding comments might overcharge the paper. In that case I would strongly recommend revising which results you want to present (and discuss) in detail, and which could be left for a supplementary information.

**Section 3.1.** I have missed a broader discussion in this part (or did you mean to have this section as strictly results only?). You mentioned that fen peat had higher initial microbial biomass, but you may go further than that. What other differences in peat properties (decomposition level, density, organic matter and water quantities, ect) or the context of the peatlands itself? Maybe permafrost has some impact, maybe climatic conditions? You present a lot of important and interesting information in this section, but you do not comment more on it. Maybe more perspective globally or within the subarctic peatlands? Why would temperature and nutrient additions/limitations affect $CO_2$ and $CH_4$ production differently?

Overall, I recommend considering connecting this section with 3.2, in which you present same information, only different units. But the missing discussion questions remain.

**Section 3.3. Lines 283-285.** I am not sure I follow your interpretation here. $CO_2$ and $CH_4$ are, in large, produced by different mechanisms and different microorganisms. Combining the production of both

these gases does not necessarily better explain the nutrient effects, but maybe a better explanation of what you meant could suffice. Nevertheless, I would recommend discussing each gas separately, taking into account different needs of methanogens and other heterotrophic microbes; also, please consider a potential effect of gas consumption by other microorganisms (e.g., methanotrophs) or chemical reactions.

**Line 294** "It is difficult to imagine that the observed CO2 and CH4 production throughout the incubation period could have occurred without microbes in fen soil samples". I understand you added this sentence following comments from previous reviewers, but I find it slightly excessive, and I would recommend removing it. The remaining explanation, as to why you do not present the results from fen, is sufficient. Alternatively, I would recommend reconsidering whether you need to present the microbial biomass changes at all (maybe move Table 3 into supplementary material?). The experiment results are informative and interesting without this part; moreover, it becomes confusing since you cannot compare fen with bog, as you do in other instances.

**Line 313.** By "the use of solid organic particles", do you mean the microbial use, or the continuous leaching of DOC from the particles you have. I recommend considering (and clearly naming) both options.

---

## Author Response (AR2)

6 February 2025

Dear Editor,

Thank you for reviewing our revised manuscript. We sincerely appreciate the thoughtful suggestions and detailed comments provided by the two Reviewers that were helpful in improving the clarity of our manuscript, and we are pleased to address the minor revisions necessary for this manuscript to be published in SOIL. We agree with the Reviewers' general and specific technical comments and suggestions and believe we have addressed all of the comments in the revised manuscript. We also thank the Topic Editor, and we hope that our revisions will allow you to recommend the manuscript for publication.

Below, we provide our point-by-point responses to the referees' suggestions and comments, with the line numbers corresponding to the marked-up manuscript version. According to the key points that you summarized in the decision letter, the title has been shortened, and the methods section has been supplemented with additional clarifications regarding the sampling and laboratory procedures. We believe that the results and discussion sections have been significantly improved by incorporating the referees' feedback. The concluding section of this manuscript has been enhanced to better summarize the findings of this study and to include remarks for future research.

Reviewer's comments are in black text. Our responses are in blue text.

Thank you for your time and consideration.

Sincerely,

Eunji Byun (On behalf of authors)
Department of Earth System Sciences
Yonsei University, Seoul, Korea

**Responses to Reviewer #1**

The authors investigate the effects of nutrient enrichment on soil CO2 and CH4 production in peatlands in Canada. This is a revised version of the manuscript. The authors answered the reviewers' questions and took their comments serious. Overall, the manuscript clearly benefited from the revision. There are still some flaws that need to be revised, but the manuscript is now well structured and clearly addresses its limitations.

We greatly appreciate your previous comments and this review and suggestions, which have improved the quality of our study. The revised version of the manuscript has been further improved by addressing the additional comments and detailed suggestions, as well as those by Reviewer#2. Kindly refer to the following for detailed responses to your comments.

See my detailed comments below (line numbers refer to the revised manuscript w/o track changes):

Title:
The title fits much better now. However, as SOIL asks the title to be "concise but informative", you might shorten it a little.

**Completed.** Thank you for your feedback. We have revised the title to "Effects of nitrogen and phosphorus amendments on $CO_2$ and $CH_4$ production in peat soils of Scotty Creek, Northwest Territories: Potential considerations for wildfire and permafrost thaw impacts on peatland carbon exchanges"

Abstract:
l. 16 I wonder if "anthropogenic emissions" is the correct expression here - as you are primarily interested in inputs into the soil.

**Completed.** Thanks for your comment. As the reviewer suggested, "anthropogenic emissions" here may be ambiguous in this context, so we have changed it to "anthropogenic inputs" in the revised manuscript [l. 17].

l. 24 You don't need "a series of" here.

**Completed.** Thanks for your suggestion. Corrected [l. 25].

l. 25 "...both...together" is redundant.

**Completed.** Thanks for noting this. We have revised the sentence to "dissolved N and P together" for clarity [l. 26].

l. 31-32 I suggest to omit the last sentence here. It is not necessary to state in the Abstract that further investigation is needed.

**Completed.** Thanks for your comment and we agree with you. We have deleted the previous version of the last sentence in the Abstract ("Our preliminary … requires further investigation.")

but instead added a different point (according to the Referee #3's suggestion) in the revised manuscript [l. 32-36].

Introduction
l. 36 I suggest "...across these regions..." - so that this part clearly refers to the subarctic regions you named before.

**Completed.** Thanks for your suggestion. Corrected in the revised manuscript [l. 40].

l. 35-37 This is still confusing. I already asked in my first review why you first name "subarctic regions" and then name western Canada, Siberia and Alaska in addition - as if these regions are something completely different. What is your point here? Do you want to express that these regions are particularly affected by wildfires? Or do you want to express that it is not only the subarctic that is affected? Please revise this sentence and clarify it.

**Completed.** Here, our aim is to connect the effects of permafrost thaw (and overlying peatland collapse) on carbon loss with the potential additional (and potentially severe) impacts of wildfires. While the permafrost thaw and peatland collapse are considered widespread phenomena across the subarctic regions (due to global climate warming), the recent occurrences and trends of wildfires are more complicated due to many factors, including hydroclimate, fuel load, and ignition source dynamics. We do not intend to say that the wildfires are increasing across the subarctic regions overall, as some parts may present different patterns in fire activity depending on how it is defined (*e.g.,* focusing on its extent, severity, frequency, and others). Thus, here we refer to these specific studies that have recently reported the increased cases of wildfire activities for each mentioned region, rather than proposing a general increase of wildfires for the entire region.

Given the comment, we have revised the sentence for clarity as follows:

 "While permafrost thaw and peatland collapse is rapidly expanding across these regions (Porter et al., 2019; Quinton et al., 2019; Hugelius et al., 2020), some parts of the northern boreal and subarctic regions, such as western Canada (Gibson et al., 2018), Siberia (Talucci et al., 2022), and Alaska (Mekonnen et al., 2022), have also experienced increased wildfire activity in recent years likely further altering the effects of permafrost thaw on soil carbon stability" [l. 39-43].

l. 45 I suggest to omit "even" and rephrase the second part of this sentence: "...with high P concentrations persisting for several years..." or something similar.

**Completed.** Thanks for your comment. As suggested, we have omitted and rephrased accordingly in the revised manuscript. "… with high P concentrations possibly persisting for several years after a fire …" [l. 51-52].

l. 57 "poor-nutrient ecosystem" is not a common term. I suggest to rephrase it: "ecosystems poor (or low) in nutrients".
**Completed.** Thank you. We have rephrased the sentence to "While peatland soils tend to be generally poor in nutrients such as N and P, the magnitude of nutrient limitation of the soil microbial community …" [l. 63-64].

l. 60 I suggest to replace "comparatively" by "in comparison" or something similar.

**Completed.** Thanks for your suggestion. Replaced in the revised manuscript [l. 66].

l. 70 "...a lont-term P inputs"?

**Completed.** Thank you. We have revised to "P supply" according to the original study's terminology [l. 77].

l. 73 It should be either "...of the peatland ecosystem there" or "...of these peatland ecosystems".

**Completed.** Thank you and we have revised the sentence to "…of the peatland ecosystem there" [l. 80].

l. 76 In this context, "material" is usually used as plural form. And I guess it should be "...as substrate".

**Completed.** Thanks. We have revised it accordingly to "organic material" and "carbon substrate" [l. 8**2**].

l. 82-83 I suggest to rewrite this sentence to better connect it with the previous one: "To address which strategy is activated, laboratory soil incubation experiments..."

**Completed.** We have revised the sentence to: "To address which strategy is activated, soil incubation experiments provide one approach to measure the changes in carbon gas production rates and net microbial biomass changes" [l. 88-90].

Methods
l. 107-110 You should add information on your sampling approach here. You only mention "shallow peat cores" that "were taken". Only in the next paragraph you mention the liners.
What type of liner did you use?

**Completed.** The liners were made of transparent plastic material from AMS, Inc. We have added this information in the Methods section of the revised manuscript [l. 115-116]. "The peat cores were collected in transparent plastic liners (3-inch inner diameter, AMS, Inc)."

l. 117 Do you mean "...to preserve initial porewater released upon..." here?
**Completed.** Yes, and this has been now corrected in the revised manuscript [l. 124].

l. 123, 124 You should not start a sentence with an abbreviation.

**Completed.** Corrected not to start with TOC, TN, or TP [l. 129-132].

l. 127 You should introduce abbreviations first (here: ICP-OES). Is there a reason you changed it during the revision?

**Completed.** Thanks for noting this. No, this was a mistake that occurred during the earlier revision. We have re-checked the whole manuscript to identify all abbreviations at their first occurrence [l. 134-135].

l. 146 No comma needed here.

**Completed.** Removed [l. 154].

l. 165-167 You changed the temperature every three days and performed gas sampling every third day after the chamber was set to a new temperature? Does this mean you performed gas sampling more or less at the same time as the temperature was changed? Please clarify.

**Completed.** Thank you for this comment: We collected the gas samples and then changed the temperature. This has been corrected in the revised manuscript to explicitly state "after" in the sentence [l. 176].

Results and Discussion
l. 250-253 Equations and approaches (including explanations) should be mentioned in the Methods section.

**Completed.** Thanks for your suggestion. We have now moved them to the Methods section [from l. 266-270 to l. 235-239].

l. 254-2554 "...except for the only N addition" sounds strange, please rephrase. Only "...except for N addition to the bog soil" should be fine.

**Completed.** Thanks for your comment. It was written this way during the previous revisions, as it may be confusing for some readers (*e.g.,* whether N addition includes the case of both N and P addition). After some additional thought, we have decided to change it according to the current suggestion, as it should be obvious enough in Table 2 [l. 272].

l. 258 Again, you don't need "N only addition" here - you introduced N addition, P addition and NP addition. This is sufficient to distinguish between the treatments.
**Completed.** Thanks for your suggestion. We have revised this in the manuscript [l. 275].

l. 314 Same here.
**Completed.** Thanks for your suggestion. We have revised this in the manuscript [l. 332].

l. 337 "...could best enhance..." sounds not very scientific. I suggest to replace best here by some expression that indicates that NP addition enhanced microbial CUE the most.

**Completed.** Thank you for this comment. The sentence has been revised, addressing the comment below [l. 355-356].

l. 337 Please avoid "...while...while..." constructions within one sentence.

**Completed.** Thanks for your comment. We have revised the sentences to: "The largest net microbial biomass growth was observed with the NP addition in the bog peat, presumably because of enhanced assimilation of DOC instead of auxiliary respiration (Giesler et al., 2011; Manzoni et al., 2012; Sinsabaugh et al., 2013; Lin et al., 2014). This is consistent with the reduced $CO_2$ production observed following the NP addition (Fig. 3), despite the larger drop in DOC relative to the N and P additions (Fig. 4). Overall, this implies that NP addition promoted the degradation of soil organic carbon that, under the anaerobic conditions in the soil environment, resulted in the higher cumulative $CH_4$ production" [l. 353-360].

Figure 4 Please take care that captions include all necessary information needed to understand the figure, this includes abbreviations.

**Completed.** Thanks for your suggestion. We have revised the caption accordingly [l. 365-367].

Table 4: Same here. Please check all captions if they include all necessary information.

**Completed.** Thank you for this comment. We have revised the captions to clarify and include necessary information [l. 380-381].

l. 371-384 Obviously, something went wrong with formatting here.

**Completed.** Thanks for noting this. Yes, it was a formatting error, which has been now corrected [l. 392-393].

Future research suggestions
I'd suggest to replace this part by a conclusion of your work that includes future research suggestions.

**Completed.** Thanks for your suggestion. In the revised manuscript, the following paragraph summarizing the main results of this study has been added, followed by suggestions for future research, and the title of the section was changed to 'Conclusions' [l. 407-413].

"This study demonstrated that the addition of dissolved N and P to short-term laboratory soil incubations causes changes in microbial C, N, and P cycling with marked differences between fen and bog peat soils. The added availability of dissolved N and P changes the temperature sensitivity ($Q_{10}$) of the soil carbon gas production rates, with an overall decrease in apparent $Q_{10}$ values, which we attribute to a compensatory effect of microbial activity under low-temperature conditions. Given the vast amount of organic-rich peat deposits in northern permafrost regions, where the landscape is rapidly thawing and increasingly experiencing wildfires, scaling up potential perturbations of increased nutrient N and P inputs and changes in nutrient ratios, as well as the long-term consequences for peatland-atmosphere carbon exchanges, will require further research."

l. 385 As far as I understand it, a pulse is something that recurs periodically - so something different than a one-time addition.

**Completed.** Thanks for your comment. We have now removed "pulsed" from the sentence in the revised manuscript. [l. 416]

l. 393 I guess with "pulsed" you mean a one-time input in contrast to the continuous input. I suggest to replace it (compare my comment to line 385) as it is too easy to be misunderstood.

**Completed.** We have revised it to "one-time" in contrast to the continuous input. [l. 424]

**Responses to Reviewer #3**

I enjoyed reading the manuscript and I found the study well-designed and the manuscript overall written well and definitely worthy consideration for publication. It still, nevertheless, requires some careful revision and (sometimes) restructuring, to better lead the readers through the complex methodology and comprehensive (hence, also confusing) results and especially discussion. I recommend the manuscript for publication, after a further revision, and I encourage the authors to complete the great task they started so well. Please see my more detailed comments below:

Thank you for your careful review and positive and constructive feedback on our study. We have carefully addressed each of your comments and made the necessary revisions to improve the manuscript as well as improved the clarity and structure of the methods, results, and particularly the discussion sections. We sincerely appreciate your insights, which have contributed to improving the quality of our work, and we hope the revisions address your concerns effectively. Please refer to the following sections for detailed responses to your comments.

Title: I find is somewhat long, even though I understand it was corrected based on previous comments. Nevertheless, it is strikingly informative, so I suppose it is mostly for the editor and the authors to decide if to leave it so long or maybe cut the last part following the colon.

**Completed.** As it was also suggested by Reviewer#1, we have carefully considered the suggestions for the revised title and decided to shorten it to: "Effects of nitrogen and phosphorus amendments on $CO_2$ and $CH_4$ production in peat soils of Scotty Creek, Northwest Territories: Potential considerations for wildfire and permafrost thaw impacts on peatland carbon exchanges"

Abstract: overall well written, well summarises the project and presents its aims. The only sentence I would recommend reconsidering is the last one – all our studies are not final, and more research can always help, but I doubt this is the only strong conclusion you can make out of this experiment. I can see that in the previous round of reviews you received some harsh criticism, which I found unmerited. Your findings have an important value and should have a corresponding concluding sentence in the end of the abstract.

**Completed.** Thanks for your comment and your positive feedback is much appreciated. We have omitted the last sentence of the abstract according to the Reviewer #1's recommendation and added a concluding sentence as: "Our results demonstrate that porewater nutrient availability and soil carbon cycling interact in complex ways to change $CO_2$ and $CH_4$ production rates in peatland soils,

with potentially far-reaching implications for the impacts of wildfires and permafrost thaw on peatland-atmosphere carbon exchanges" in the revised manuscript [Lines 34-36].

Introduction: overall good, I only have one comment related to Table 1 and the style of presenting the references
Line 45. I would contest the need of the Table 1 to prove your point here, and I would recommend moving the table to supplementary materials. While the table shows important examples of nutrient enrichment after wildfires, it is sightly out of scope for your study's objectives, and I doubt you must have it there, in the main text, to justify your experiment. You are doing an excellent justification with your introduction text already. I would rather focus on more clear presentation of the references for the first sentences of this paragraph.

**Completed.** We appreciate this point and have revised the related sentence for a better connection with the references by removing the phrase to connect in with Table 1 [Line 52]. Additionally, the table is now moved to the Methods section [Line 164] to help understanding of the experiment procedure as well as to be more relevant to the purpose of this table.

This comment is applicable for the entire text – the reader is often left with lengthy stretches of text without references, followed with a bunch of references in one place (e.g., Emelko et al., 2016; Van Beest et al., 2019; Emmerton et al., 2020; Orlova et al., 2020). In such cases, readers might be left confounded – do all these mentioned studies prove one statement (as it is in line 50; in which case, it is not necessary to show that many references, 1-3 is usually enough) or all 3 to 5 statements before that (as in lines 45, 80, etc… in which case, it would be beneficial to attribute the references to their corresponding statements).

**Completed.** Thank you for your comment and making this point to improve the manuscript. We believe the revised manuscript has been improved by rephrasing those lengthy statements and adding more context to the references cited, for example, "While permafrost thaw and peatland collapse is rapidly expanding across these regions (Porter et al., 2019; Quinton et al., 2019; Hugelius et al., 2020), some parts of the northern boreal and subarctic regions, such as western Canada (Gibson et al., 2018), Siberia (Talucci et al., 2022), and Alaska (Mekonnen et al., 2022), have also experienced increased wildfire activity in recent years likely further altering the effects of permafrost thaw on soil carbon stability. For example, …" [l. 39-43].

Methods. It was detailed enough and mostly easy to follow. Adding a graphic summary or a table pointing to each step done before and during the experiment would greatly help readers, but I would not insist on adding one, it is just an idea.

**Completed.** Thank you for your suggestion. We have made every effort to describe the experimental steps as clearly as possible.

Section 2.1. I recommend adding some essential information that would help readers to make better interpretations of the results. Firstly, you mention that the study site is within the discontinuous permafrost area. It would be important to indicate whether the two peatlands you study are affected by permafrost, and most importantly, whether the coring sites are underlain by

permafrost. Secondly, you sampled in October, so it would be worth mentioning if the peat samples were already frozen when you sampled, and if they remained frozen during the travel to Waterloo.

**Completed.** The peat soils cored were not frozen at the time of sampling, as shown in the photos of the sampling borehole in Figure 1. We stored the samples frozen in the laboratory freezer until we were able to start the incubation experiment. The description of peatland types as thermokarst bog and channel fen indicates the thawed status of permafrost directly underneath the sites, although there could have been remnant permafrost in surrounding areas, given the discontinuous nature of the study region. To better reflect this, we have slightly modified the related sentence to "These peat cores were transported to the University of Waterloo, Waterloo, Canada, and then stored in a -20°C freezer until being thawed to start the experiment" [Lines 116-117].

Section 2.2. Line 121. Was the bulk density used anywhere in the study? Judging from the method you used to estimate it, it might not be the best variable to present. As I did not find any further mentions of density in the manuscript, I would recommend removing it from the methods as well – it slightly reduces the quality of the study, when mentioned like that. But if you used density somewhere else, then keep it.

**Completed.** Thanks for your comment and suggestion. For completeness, the bulk density values are given in the related dataset of this article.

Results and Discussion. I would recommend a detailed revision of this part. You have a good study design and diverse data to present. The latter often causes difficulties, when writing a clear discussion, and this was the main issue I identified while reading your manuscript. If it helps, you may consider separating results from discussion; otherwise, please carefully revise each section, and make sure you do not limit yourself to commenting on each particular detail of presented results but also expand into a larger context. In many cases, a presentation of what is normally expected for different variables would make the discussion clearer (not everyone reading the article will be as knowledgeable in all the domains you touch). Overall, I missed a discussion on the meaning of your findings – what did we learn from this, where do we go here? May we expect similar wildfire effects in other subarctic wetlands: what will be the effect of permafrost disappearance, or increasing peatland ground temperatures? Please, do not get discouraged by the previous reviewer's comments about your experiment being unrepresentative. While it is admirable that you carefully acknowledge the limitations of the study, we often must work with what we have, and careful interpretations (based on similar studies) are acceptable. P.S., fens and bogs are highly representative for boreal regions and comparing these two is a significant contribution to the scientific knowledge, even if you work with only one particular study site. Boreal/subarctic regions are strikingly similar across North America and Eurasia.

**Completed.** Thank you for your valuable feedback. While we understand the benefit of separating the Results and Discussion sections, we believe that combining them into a single section is more suitable for the structure of this short communication article. This format allows us to present the experimental results alongside the immediate insights they provide, making the connection between the findings and their implications more straightforward for the reader, who may not be familiar with this type of experiment.

We acknowledge that presenting the expected trends for different variables and their resulting effects could help readers better understand the new insights from this experiment. However, relatively few studies are available to clearly define the expected trends for each variable we discuss hitherto. We believe that a review article could be developed in the future to address these points while continuing laboratory experiments to further investigate the suggested effects from our experiment.

The additional paragraph [Lines 408-414] in the Conclusions section (previously titled 'Future research suggestions') in the revised manuscript summarizes the main findings of this study, as well as the additional last sentence in the abstract [Lines 34-36]. Although we much appreciate the suggestion to make more broad statements for the study's insights, we are cautious about making such statements in this article, given the concerns raised by previous reviewers and the topic editor. Still, we believe that interested researchers will gain valuable and broad enough insights for future research directions to investigate the important peatlands in boreal/subarctic regions.

I am aware that you have submitted for a short communication, so my preceding comments might overcharge the paper. In that case, I would strongly recommend revising which results you want to present (and discuss) in detail, and which could be left for supplementary information.

**Completed.** Thank you for this advice. We have carefully revisited our manuscript with the two Reviewers' comments and suggestions in mind.

Section 3.1. I have missed a broader discussion in this part (or did you mean to have this section as strictly results only?). You mentioned that fen peat had higher initial microbial biomass, but you may go further than that. What other differences in peat properties (decomposition level, density, organic matter and water quantities, ect) or the context of the peatlands itself? Maybe permafrost has some impact, maybe climatic conditions? You present a lot of important and interesting information in this section, but you do not comment more on it. Maybe more perspective globally or within the subarctic peatlands? Why would temperature and nutrient additions/limitations affect $CO_2$ and $CH_4$ production differently?
Overall, I recommend considering connecting this section with 3.2, in which you present same information, only different units. But the missing discussion questions remain.

**Completed.** Thanks for your comment and suggestion. The section 3.1 is intended to present the results of gas flux measurements as clearly as possible, despite the complexity of data we obtained (raw data are found in the related dataset). The context of adding nutrients for measuring the potential differences in resulting carbon gas production rates has been demonstrated in the Introduction section. However, there is no study to specifically associate with the current results as the experimental setting was different. Thus, the questions raised by this comment are not possible to be confidently addressed based on our relatively small-sample measurement results. We agree with connecting this section with 3.2 and the changes have been made in the revised manuscript [Line 264].

Section 3.3. Lines 283-285. I am not sure if I follow your interpretation here. $CO_2$ and $CH_4$ are, in large, produced by different mechanisms and different microorganisms. Combining the production of both these gases does not necessarily better explain the nutrient effects, but maybe

a better explanation of what you meant could suffice. Nevertheless, I would recommend discussing each gas separately, taking into account different needs of methanogens and other heterotrophic microbes; also, please consider a potential effect of gas consumption by other microorganisms (e.g., methanotrophs) or chemical reactions.

As the microbial production of $CO_2$ and $CH_4$ are influenced by various soil internal factors, including oxygen availability, we were not able to monitor all the detailed factors for the incubation of multiple jars at the same time with a limited amount of sample. Also, the $CH_4$ gas could be substrate for $CO_2$ production internally by methanotrophs as you mentioned, and thus the measurement of $CH_4$ at the certain time point is not fully representative the gross rates of $CH_4$ production but the net results over the three days in a new temperature. Therefore, we summarized the net loss of soil organic carbon through both gas production, which we think is more representative for the cumulative effects of added nutrients in overall microbial activity. To demonstrate the details about the microbial community dynamics, additional DNA extraction measurements will be necessary to monitor the potential shifts in dominant communities over time and with nutrient effects.

Line 294 "It is difficult to imagine that the observed CO2 and CH4 production throughout the incubation period could have occurred without microbes in fen soil samples". I understand you added this sentence following comments from previous reviewers, but I find it slightly excessive, and I would recommend removing it. The remaining explanation, as to why you do not present the results from fen, is sufficient. Alternatively, I would recommend reconsidering whether you need to present the microbial biomass changes at all (maybe move Table 3 into supplementary material?). The experiment results are informative and interesting without this part; moreover, it becomes confusing since you cannot compare fen with bog, as you do in other instances.

**Completed.** Thanks for your comment and suggestion, we have removed them in the revised manuscript [Lines 311-312].

Line 313. By "the use of solid organic particles", do you mean the microbial use, or the continuous leaching of DOC from the particles you have. I recommend considering (and clearly naming) both options.

**Completed.** While we cannot directly know which process was dominant in the use of solid organic particles, we assume both processes were involved. The sentence has been revised to suggest both possible processes in the parentheses "…production of DOC from particulate, either through microbial enzymatic decomposition or chemical leaching" [Lines 330-331].